# From Shapes to Shapes: Inferring SHACL Shapes for Results of SPARQL CONSTRUCT Queries

## ABSTRACT

SPARQL CONSTRUCT queries allow for the specification of data processing pipelines that transform given input graphs into new output graphs. It is now common to constrain graphs through SHACL shapes allowing users to understand which data they can expect and which not. However, it becomes challenging to understand what graph data can be expected at the end of a data processing pipeline without knowing the particular input data: Shape constraints on the input graph may affect the output graph, but may no longer apply literally, and new shapes may be imposed by the query template. In this paper, we study the derivation of shape constraints that hold on all possible output graphs of a given SPARQL CONSTRUCT query. We assume that the SPARQL CONSTRUCT query is fixed, e.g., being part of a program, whereas the input graphs adhere to input shape constraints but may otherwise vary over time and, thus, are mostly unknown. We study a fragment of SPARQL CONSTRUCT queries (SCCQ) and a fragment of SHACL (Simple SHACL). We formally define the problem of deriving the most restrictive set of Simple SHACL shapes that constrain the results from evaluating a SCCQ over any input graph restricted by a given set of Simple SHACL shapes. We propose and implement an algorithm that statically analyses input SHACL shapes and CONSTRUCT queries and prove its soundness and complexity.

## CCS CONCEPTS

• **Information systems** → *Graph-based database models*; **Resource Description Framework (RDF)**; *Query languages*; *Extraction, transformation and loading*.

## KEYWORDS

SHACL, semantic queries, SPARQL CONSTRUCT, data pipelines

**ACM Reference Format:**
Anonymous Author(s). 2018. From Shapes to Shapes: Inferring SHACL Shapes for Results of SPARQL CONSTRUCT Queries. In *Proceedings of Make sure to enter the correct conference title from your rights confirmation emai (Conference acronym 'XX)*. ACM, New York, NY, USA, 19 pages. https://doi.org/XXXXXXX.XXXXXXX

## 1 INTRODUCTION

Shape description languages like SHACL [15] can play two different, but equally important roles. They can be used *normatively* in order to impose schematic constraints on the evolution of a graph, such that a triple store may automatically reject illegitimate configurations. They can also be used *informatively* such that a software developer knows how to display the graph, or to inform downstream applications, e.g., [17].

Graph query languages like SPARQL CONSTRUCT or G-CORE [1] allow for the fruitful composition of queries into data processing pipelines. To query a given pipeline, the developer must understand what it may output, regardless of its inputs. Even if the possible inputs to a query or composition of queries are well-described using a shape language like SHACL, it becomes very challenging to understand which shape constraints apply after one or several querying steps. SHACL constraints that apply on the input graph may or may no longer apply, e. g., existential quantification may become inapplicable because the corresponding relationship might not be part of the WHERE clause, and new constraints may or may not be imposed by the CONSTRUCT template. A developer may hold misconceptions about such structural concerns of the result graph, which might even seem to be endorsed by one particular graph instance, but can lead to errors (e.g., when processing query results within a program) for other valid instances of input graphs.

In this paper, we define the problem of computing a set of SHACL shapes characterizing the possible output graphs of a SPARQL CONSTRUCT query based on (1) the set of shapes applicable to input graphs, and (2) the graph patterns and the template of the query. We present an algorithm for constructing a sound upper approximation by statically analyzing shapes and query, relying on an encoding in description logics, and *without* referring to any specific input graph. Thus, our approach allows for investigating SPARQL CONSTRUCT queries and data processing pipelines regardless of what valid data will be encountered in the future.

*Outline.* The remainder of this paper is structured as follows. In Section 2 we introduce foundations, including a subset of SPARQL queries, SHACL shapes and the fragment of description logics we rely upon. In Section 3 we formalize our validation problem. Throughout Section 4, Section 5 and Section 6 we break the validation problem down into subproblems, and present algorithms for solving them. In Section 7 we introduce a preliminary implementation of our approach, present related work in Section 8, and finally conclude in Section 9. The *appendix* contains full proofs (including a proof for NP-hardness of our approach), extended examples, experimental results showing feasibility of the implementation, and details on how the approach can be generalized to a larger fragment of SHACL.

## 2 FOUNDATIONS

We use uppercase letters $A, B, E \in \mathbf{C}$ for description logic *concept names*, lowercase letters $a, b, e \in \mathbf{I}$ for description logic *individual names*, and lowercase letters $p, r \in \mathbf{R}$ for description logic *role names*. We interpret all RDF classes, RDF instances and RDF properties as

description logic concepts, individuals, and roles, respectively. We use lower case letters $w, x, y, z \in \mathbf{V}$ as SPARQL variables. We assume that $\mathbf{C}, \mathbf{I}, \mathbf{R}$, and $\mathbf{V}$ are four *finite*, pairwise disjoint sets. While we use finite sets to simplify definitions, this is not a restriction since for practical applications, these sets can be as large as needed.

For clarity, we always use description logic terminology rather than mixing it with RDF and SPARQL terminology, e.g., we will refer to "concept" rather than "RDF class".

## 2.1 The Description Logic $\mathcal{ALCHOI}$

We use the description logic $\mathcal{ALCHOI}$ to define the semantics of RDF graphs and SHACL shapes following the formalism by Bogaerts et al. [6]. We next present the standard $\mathcal{ALCHOI}$ syntax and assume standard semantics as defined in Baader et al. [4].

**Definition 1** ($\mathcal{ALCHOI}$ concept descriptions). $\mathcal{ALCHOI}$ *concept descriptions* are defined by the following grammar

$$C ::= \top \mid \bot \mid A \mid \neg C \mid \{a\} \mid C \sqcap C \mid C \sqcup C \mid \exists \rho.C \mid \forall \rho.C$$
$$\rho ::= p \mid p^-$$

where the symbols $\top$ and $\bot$ are two special concept names, and $A$, $a$, and $p$ stand for concept names, individual names, and role names, respectively. Given two concept descriptions $C$ and $D$, two individual names $a, b \in \mathbf{I}$, and two role descriptions $\rho_1, \rho_2$ (as defined above), $C \sqsubseteq D$ and $\rho_1 \sqsubseteq \rho_2$ are *axioms*, $a{:}C$ is a *concept assertion* and $(a, b){:}p$ is a *role assertion*. We write $C \equiv D$ as an abbreviation for two axioms $C \sqsubseteq D$ and $D \sqsubseteq C$, and likewise for $\rho_1 \equiv \rho_2$.

## 2.2 Simple RDF Graphs

According to the RDF specification [28], an RDF graph is a finite set of triples whose elements belong to three pairwise disjoint sets: IRIs, blank nodes, and literals. For convenience, we assume the fragment of RDF graphs, called *Simple RDF graphs*, that only considers triples whose elements are IRIs. Furthermore, we assume that IRIs are partitioned in the four sets $\mathbf{C}, \mathbf{I}, \mathbf{R}$, and $\{\text{rdf:type}\}$, and that an RDF triple has either the form $(a, p, b)$ or $(a, \text{rdf:type}, A)$ where $a, b \in \mathbf{I}$ and $A \in \mathbf{C}$. We interpret each triple $(a, p, b)$ as an assertion $(a, b){:}p$, and each triple $(a, \text{rdf:type}, A)$ as an assertion $a{:}A$. With these assumptions we define Simple RDF graphs, and introduce running examples in Figure 1.

**Definition 2** (Simple RDF Graph Syntax). A *Simple RDF graph* (or just *graph*) is an $\mathcal{ALCHOI}$ ABox $G$ where the concept description of each concept assertion in $G$ is a concept name $A \in \mathbf{C}$.

Bogaerts et al. [6] highlight that RDF graphs have two different semantics, depending on the inference task we want to perform: If the task is *deduction*, the semantics of a graph is given by an ABox, and following the no-unique-name, no-domain-closure and open-world assumptions. If the task is *validation*, the semantics is given by a model. Instead of relying on a model-theoretic semantics for validation, our approach benefits from a proof-theoretic semantics. As Reiter [23] suggests, the model theoretic semantics of databases can be defined in proof theoretic terms: A database can be seen as a set of formulas instead of a model, where queries are formulae to be proven, and satisfaction of constraints is defined in terms of consistency. We can therefore extend the deduction semantics of Simple RDF graphs with axioms that encode these assumptions,

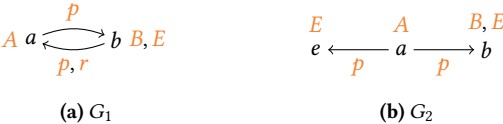

**(a)** $G_1$      **(b)** $G_2$

**Figure 1: Two example graphs, where we visualize rdf:type edges as floating labels next to nodes (e.g., $A$ $a$ for $a{:}A$).**

which are based on the proof-theoretic semantics for relational databases by Reiter [23].

Proposition 1 below implies the equivalence of the Bogaerts et al. [6] model theoretic SHACL semantics (Definition 3) and our proof theoretic SHACL semantics (Definition 4).

**Definition 3** (Graph Interpretation [6]). The *canonical interpretation* of a Simple RDF graph $G$ is the interpretation $\mathcal{I}_G$ such that $\Delta^{\mathcal{I}_G} = \mathbf{I}$; for each $a \in \mathbf{I}$, $a^{\mathcal{I}_G} = a$; for every concept name $A \in \mathbf{C}$, $A^{\mathcal{I}_G} = \{a \mid a{:}A \in G\}$; and for every role name $r \in \mathbf{R}$, $r^{\mathcal{I}_G} = \{(a, b) \mid (a, b){:}r \in G\}$. A graph $G$ is *model-valid* according to a set $\Sigma$ of $\mathcal{ALCHOI}$ axioms if and only if $\mathcal{I}_G$ is a model of $\Sigma$.

**Definition 4** (Simple RDF Graph Validation Semantics). The axioms of a Simple RDF graph $G$, denoted $\mathcal{T}_G$, are the TBox consisting of the following $\mathcal{ALCHOI}$ axioms:

(1) *Domain Closure Assumption (DCA)*: $\top \equiv \bigsqcup_{a \in \mathbf{I}} \{a\}$.
(2) *Unique Name Assumption (UNA)*: $\{a\} \sqcap \{b\} \equiv \bot$, for each pair of distinct individual names $a, b \in \mathbf{I}$.
(3) *Closed-World Assumption (CWA)*:
   - $A \equiv \bigsqcup_{a:A \in G} \{a\}$, for each concept name $A \in \mathbf{C}$,
   - $\exists p.\{a\} \equiv \bigsqcup_{(b,a):p} \{b\}$, and
   - $\exists p^-.\{a\} \equiv \bigsqcup_{(a,b):p} \{b\}$, for each role name $p \in \mathbf{R}$ and each individual name $a \in \mathbf{I}$.

$(\mathcal{T}_G, G)$ is the *validation knowledge base* of $G$. A graph $G$ is *proof-valid* according to a set $\Sigma$ of $\mathcal{ALCHOI}$ axioms if and only if $\Sigma$ is consistent with the validation knowledge base of $G$ (i.e., the knowledge base $(\mathcal{T}_G \cup \Sigma, G)$ admits a model).

**Proposition 1.** *For a graph $G$ and set of $\mathcal{ALCHOI}$ axioms $\Sigma$, the following statements are equivalent: (i) $G$ is model-valid according to $\Sigma$, (ii) $G$ is proof-valid according to $\Sigma$, and (iii) $G$ is proof-valid according to $\{\varphi\}$ for every $\varphi \in \Sigma$.*

## 2.3 Simple SHACL Shapes

Following the idea that a SHACL schema is a description logic TBox [6], a SHACL shape is an axiom of the form $\psi \sqsubseteq \phi$ where $\psi$ and $\phi$ are concept descriptions, called the *target query* and the *shape constraint*, respectively. We next define the core fragment of SHACL we consider here. (For an extension to $\mathcal{ALCHOI}$ constraints, see the *appendix*.)

**Definition 5** (Simple SHACL Syntax). A *Simple SHACL shape* (or just *shape*) is an $\mathcal{ALCHOI}$ axiom $\psi \sqsubseteq \phi$ such that the concept expressions $\psi$ and $\phi$ are defined by:

$$\psi ::= A \mid \exists p.\top \mid \exists p^-.\top$$
$$\phi ::= A \mid \exists p.A \mid \forall p.A \mid \exists p^-.A \mid \forall p^-.A$$

A *Simple SHACL schema* $\mathcal{S}$ is an $\mathcal{ALCHOI}$ TBox that consists of a finite set of Simple SHACL shapes.

Given that shapes are defined in terms of $\mathcal{ALCHOI}$ axioms, their semantics is defined in terms of the semantics of $\mathcal{ALCHOI}$ axioms over the validation knowledge base of a graph.

**Definition 6** (Simple SHACL Semantics). A graph $G$ is *valid* for a set $S$ of Simple SHACL shapes, denoted valid($G, S$), if and only if $G$ is proof-valid according to $S$.

**Example 1.** Consider the set of shapes $S_1 = \{s_1, s_2, s_3\}$ where $s_1 = A \sqsubseteq \exists p.B$, $s_2 = \exists r.\top \sqsubseteq B$ and $s_3 = B \sqsubseteq E$. Shape $A \sqsubseteq \exists p.B$ targets all individuals that are instances of $A$, and requires that there exists at least one edge $p$ to a $B$. Both graphs in Figure 1 are *valid* with respect to $S_1$.

## 2.4 Simple Conjunctive CONSTRUCT Queries

This section defines the fragment of SPARQL CONSTRUCT queries this paper considers, called *Simple Conjunctive* CONSTRUCT *Queries* (SCCQ, or just *queries*). This fragment follows the semantics proposed by Kostylev et al. [16] and is restricted to basic graph patterns generated by adding variables for individual names on Simple RDF graphs.

**Definition 7** (SCCQ Syntax). An *atomic pattern* $t$ is defined by the following grammar:

$$t ::= a{:}A \mid x{:}A \mid (a, b){:}p \mid (x, a){:}p \mid (a, x){:}p \mid (x, y){:}p$$

where $A$ stands for concept names, $a$ and $b$ for individual names, $p$ for role names, and $x$ and $y$ for variables. A finite set of atomic patterns is a *basic graph pattern*. Given a basic graph pattern $P$, we write var($P$) and ind($P$) to denote the respective sets of variables and individual names occurring in pattern $P$. Given two basic graph patterns $P$ and $H$, where var($H$) ⊆ var($P$), the expression $H \leftarrow P$ is a SCCQ, where $H$ and $P$ are called the *template* and the *pattern* of the query, respectively.

A *valuation* of a basic graph pattern $P$ is a function $\mu : \mathbf{V} \cup \mathbf{I} \rightarrow \mathbf{I}$ such that $\mu(a) = a$ for every $a \in \mathbf{I}$. In a slight abuse of notation, given two elements $u, v \in \mathbf{V} \cup \mathbf{I}$ and a basic graph pattern $P$, we write $\mu(u{:}A) = \mu(u){:}A$, $\mu((u, v){:}p) = (\mu(u), \mu(v)){:}p$, and $\mu(P) = \{\mu(t) \mid t \in P\}$. Intuitively, a valuation substitutes variables in a pattern by individual names. The semantics of SCCQ is defined below.

**Definition 8** (SCCQ Semantics). The result of evaluating a SCCQ $H \leftarrow P$ over a Simple RDF graph $G$ is the Simple RDF graph, denoted $[\![H \leftarrow P]\!]_G$, defined as follows:

$$[\![H \leftarrow P]\!]_G = \bigcup_{\mu(P) \subseteq G} \mu(H).$$

Intuitively, the pattern $P$ retrieves valuations $\mu$ such that $\mu(P)$ is a subgraph of $G$, which are used to generate the output graph by replacing variables in the template.

**Example 2.** Let $q_1 = H \leftarrow P =$

$$\{y{:}E, z{:}B, (y, z){:}p\} \leftarrow \{(w, y){:}p, y{:}B, (x, z){:}p, z{:}E\}$$

. For evaluation over the first example graph, $[\![q_1]\!]_{G_1}$, we need to find valuations $\mu$ where $\mu(P) \subseteq G_1$. This holds for $\mu$ where $\mu(w) = a$, $\mu(x) = a$, $\mu(y) = b$, and $\mu(z) = b$. Hence, the result is the graph

$[\![H \leftarrow P]\!]_{G_1} = \mu(H) = \{b{:}E, b{:}B, (b, b){:}p\}$. Similarly, evaluation $[\![q_1]\!]_{G_2} = \{b{:}E, b{:}B, e{:}B, (b, b){:}p, (b, e){:}p\}$.

## 3 FORMAL PROBLEM STATEMENT

We aim to construct a set of shapes characterizing the possible result graphs of a query where the input is constrained by a set of shapes as well.

**Definition 9** (Input and Output Graph). A graph $G_{in}$ is an *input graph* with respect to a finite set of shapes $\mathcal{S}_{in}$ if valid($G_{in}, \mathcal{S}_{in}$). A graph $G_{out}$ is an *output graph* for a query $q$ and a finite set of shapes $\mathcal{S}_{in}$ if there exists an input graph $G_{in}$ such that $G_{out} = [\![q]\!]_{G_{in}}$.

**Definition 10** (Vocabulary). A *vocabulary* is the set of concept and role names that occur in a concept description $C$, shape $s$, graph $G$, or template of a query $q$, denoted voc($C$), voc($s$), voc($G$), or voc($q$), respectively.

**Definition 11** (Relevancy). Shape $s = \psi \sqsubseteq \phi$ is *relevant* for query $q$ if there exists a graph $G_+$ with voc($G_+$) ⊆ voc($q$) such that valid($G_+, \{s\}$) and $(\mathcal{T}_{G_+}, G_+) \not\models \psi \equiv \bot$, and a graph $G_-$ with voc($G_-$) ⊆ voc($q$) such that not valid($G_-, \{s\}$).

Problem OUTPUTSHAPES formalizes the set of shapes that best characterize the possible output graphs of a SCCQ. The first restriction on the solution ensures only relevant shapes are in the output, i.e., shapes that validate some graphs in the vocabulary voc($q$), but not all of them (Definition 11). This excludes, for example, shapes with targets outside the vocabulary (which are thereby vacuously satisfied), or shapes with constraints requiring concept or role names outside the vocabulary, which can never be satisfied. The second restriction states that $\mathcal{S}_{out\text{-}opt}$ defines an upper bound for the set of output graphs, while the third requires this upper bound to be minimal.

---

**Problem** OUTPUTSHAPES : $(\mathcal{S}_{in}, q) \mapsto \mathcal{S}_{out\text{-}opt}$

**Input** A finite set of shapes $\mathcal{S}_{in}$ and a SCCQ $q$.
**Output** A set of shapes $\mathcal{S}_{out\text{-}opt}$ such that:
    1. every $s \in \mathcal{S}_{out\text{-}opt}$ is relevant for $q$,
    2. for every $G$ with valid($G, \mathcal{S}_{in}$) and $G_{out} = [\![q]\!]_G$, valid($G_{out}, \mathcal{S}_{out\text{-}opt}$),
    3. the set of graphs $G$ such that valid($G, \mathcal{S}_{out\text{-}opt}$) is minimal.

---

**Example 3.** Consider $q_1$ (Example 2) and $S_1$ (Example 1). The shapes $E \sqsubseteq \exists p.B, E \sqsubseteq B \in S_{1\text{-}out}$ constrain the results of evaluating $q_1$ on *any* graph that is valid with respect to $S_1$, e.g., the example graphs in Figure 1. Shape $E \sqsubseteq \exists p.B \in S_{1\text{-}out}$ follows directly from the query template, whereas shape $E \sqsubseteq B$ is only contained in $S_{1\text{-}out}$ because $B \sqsubseteq E$ holds on all input graphs and we can thus infer that all bindings for $y$ are also bindings for $z$.

Simple SHACL shapes are not sufficiently expressive to rule out *all* impossible output graphs of a query. For example, we know for $q_1$ and $S_1$ that each instance of $E$ has a $p$ edge to itself. Simple SHACL shapes cannot express reflexiveness, so graphs without reflexive $p$ cannot be ruled out.

## 4 COMPUTING CANDIDATE OUTPUTSHAPES

We break down Problem OUTPUTSHAPES into two subproblems: The generation of a finite set of candidate shapes $\mathcal{S}_{\text{can}}$ – a superset of the solution – and the filtering of this set (Problem IsOUTPUTSHAPE).

---

**Problem 2** IsOUTPUTSHAPE : $(\mathcal{S}_{\text{in}}, q, s) \mapsto \{\text{YES}, \text{NA}\}$

**Input** A finite set of shapes $\mathcal{S}_{\text{in}}$, a SCCQ $q = H \leftarrow P$, and a shape $s$ that is relevant for this query $q$.

**Output** Does $\text{valid}(\llbracket q \rrbracket_{G_{\text{in}}}, \{s\})$ hold for every graph $G_{\text{in}}$ where $\text{valid}(G_{\text{in}}, \mathcal{S}_{\text{in}})$?

---

Algorithm 1 outlines this approach, by referring to Problem IsOUTPUTSHAPE. In Section 6, we will define a sound, but not complete, algorithm solving this problem (Algorithm 2). Thus, Algorithm 1 is a sound approximation of problem OUTPUTSHAPES satisfying its first two, but not the third condition. In the following we use $\mathcal{S}_{\text{out}}$ to refer to such an approximation of $\mathcal{S}_{\text{out-opt}}$.

---

**Algorithm 1** OUTPUTSHAPES : $(\mathcal{S}_{\text{in}}, q) \mapsto \mathcal{S}_{\text{out}}$

**Input** A finite set of shapes $\mathcal{S}_{\text{in}}$ and a SCCQ $q$.
**Output** The set of output shapes $\mathcal{S}_{\text{out}}$.
1: $\mathcal{S}_{\text{out}} \leftarrow \emptyset$, $\mathcal{S}_{\text{can}} \leftarrow$ the finite set of shapes over $\text{voc}(q)$
2: **for all** $s \in \mathcal{S}_{\text{can}}$ **do**
3:     **if** IsOUTPUTSHAPE$(\mathcal{S}_{\text{in}}, q, s) = $ YES **then**
4:         $\mathcal{S}_{\text{out}} \leftarrow \mathcal{S}_{\text{out}} \cup \{s\}$
5: **return** $\mathcal{S}_{\text{out}}$

---

In order to obtain a finite set of candidates $\mathcal{S}_{\text{can}}$, Proposition 2 allows us to discard shapes that do not describe output graphs and limit thus the search space of Algorithm 1 to the shapes that are built from the vocabulary of the query.

**Proposition 2.** *If a shape $s = \psi \sqsubseteq \phi$ is relevant for a SCCQ $q$, then $s$ satisfies one of the following two conditions: (i) $\text{voc}(s) \subseteq \text{voc}(q)$, or (ii) $\text{voc}(\psi) \subseteq \text{voc}(q)$ and $\phi$ is either $\forall p.A$ or $\forall p^-.A$ where $p \in \text{voc}(q)$ and $A \notin \text{voc}(q)$.*

In order to cover all relevant shapes that satisfy condition (i), we can include the finite combinations of elements in the vocabulary of the query. Condition (ii) requires special care: Each role name $p \in \text{voc}(q)$ defines a family of shapes of the form $\psi \sqsubseteq \forall p.A$ or $\psi \sqsubseteq \forall p^-.A$, where $A \notin \text{voc}(q)$. To explore this family, it suffices to consider a representative by including in the set of candidate shape constraints for each role name $p \in \text{voc}(q)$ the two concept descriptions $\forall p.A$ and $\forall p^-.A$, such that $A \notin \text{voc}(q)$.

The search space is therefore bounded by the vocabulary of the query, which is relatively small. In the *appendix* we show that there are $(n + 2m)(n + 4nm + 2m) - n$ candidate shapes if $\text{voc}(q)$ contains $n$ concept names and $m$ role names.

## 5 AXIOMATIZATIONS OVER QUERY EXECUTIONS

A query $q = H \leftarrow P$ works on any input graph $G_{\text{in}}$ defined by $\mathcal{S}_{\text{in}}$ (Definition 9) and returns a result graph $G_{\text{out}}$ in two steps: By matching $P$ with $G_{\text{in}}$, determining valuations $\mu$ where $\mu(P) \subseteq G_{\text{in}}$, and then by replacing variables in $H$ with these valuations producing $G_{\text{out}}$. We now want to axiomatize how all possible $G_{\text{in}}$ are connected with their corresponding $G_{\text{out}}$.

Virtually putting these axiomatizations together creates an *extended graph* that holds axioms from these two steps allowing us to prove statements about $G_{\text{out}}$. Thereby, we distinguish inputs and step outcomes by a syntactic trick that rewrites input symbols $A, p$ into *fresh* symbols $\dot{A}, \dot{p}$ after the first step, and into $\ddot{A}, \ddot{p}$ after the second step. We also write, e.g., $\dot{G}$, meaning substitution of all symbols $A, p$ in graph $G$ with $\dot{A}, \dot{p}$, respectively.

Therefore, these rewritten symbols allow us to encode assertions that are valid for only specific states of query execution. Variable bindings, on the other hand, hold throughout: We codify a variable binding $\mu(x) = a$ as a concept assertion $a{:}V_x$, where $V_x$ is a fresh concept name. Note, that we assume that all concept names and role names with dots, as well as concept names for variable concepts, exist as fresh names in $\mathbf{C}$ and $\mathbf{R}$.

Example 4 illustrates the construction of such an extended graph, which is defined in Definition 12.

**Definition 12** (Extended Graph). Given an input graph $G_{\text{in}}$ and a query $H \leftarrow P$, the following graphs are defined with correspondences to the query execution steps:

(1) The *intermediate graph* $G_{\text{med}} := \bigcup_{\mu(P) \subseteq G_{\text{in}}} \mu(P)$.
(2) The *variable concept graph* $G_{\mathbf{V}}$ containing an assertion $a{:}V_x$ if and only if there exists a valuation $\mu$ such that $\mu(P) \subseteq G_{\text{in}}$ and $\mu(x) = a$.
(3) The *output graph* $G_{\text{out}} := \llbracket q \rrbracket_{G_{\text{in}}}$.
(4) The *extended graph* $G_{\text{ext}} := G_{\text{in}} \cup \dot{G}_{\text{med}} \cup G_{\mathbf{V}} \cup \ddot{G}_{\text{out}}$.

**Example 4.** Consider $q_1$ (Example 2), $S_1$ (Example 1), and the graph $G_1$ (Figure 1) as one possible input graph for $q_1$. The respective extended graph and its components are given in Figure 2. Note, that these graphs satisfy different axioms (in different namespaces), e.g., $\exists \dot{p}^-.\top \sqsubseteq \dot{E}$ is valid in $\dot{G}_{\text{med}}$ but $\exists p^-.\top \sqsubseteq E$ is not valid in $G_{\text{in}}$. A range of axioms are valid for $G_{\text{ext}}$, such as $\dot{E} \sqsubseteq E$ or $V_y \sqsubseteq V_z$. Indeed, these axioms are valid on *every* extended graph of $q_1$, as long as $\text{valid}(G_{\text{in}}, S_1)$, e.g., $G_{\text{in}} = G_1$ or $G_{\text{in}} = G_2$ (Figure 1).

Assertions added per step are sound, but not sufficient to fully characterize what happens at each query execution step. Therefore, axioms we can find to characterize the relationships between $G_{\text{in}}$ and $G_{\text{out}}$ will be sound but incomplete. In the following sections, we will introduce additional axioms per step to extend possible inferences and thus determine a tighter description by output shapes.

Proposition 3 shows that axioms valid on any of the graphs $G_{\text{in}}$, $G_{\text{med}}$, $G_{\mathbf{V}}$ and $G_{\text{out}}$ are valid on the extended graph when applying syntactic rewriting, and vice versa.

**Proposition 3.** *Given a graph $G_{\text{in}}$ and a query $q$, let the graphs $G_{\text{med}}$, $G_{\text{out}}$, and $G_{\text{ext}}$ be defined according to Definition 12. For every axiom $\varphi$ that does not include names with dots (e.g., $\dot{A}$, $\ddot{A}$, $\dot{p}$, $\ddot{p}$), the following equivalences hold:*

(1) $\text{valid}(G_{\text{in}}, \{\varphi\})$ *if and only if* $\text{valid}(G_{\text{ext}}, \{\varphi\})$.
(2) $\text{valid}(G_{\text{med}}, \{\varphi\})$ *if and only if* $\text{valid}(G_{\text{ext}}, \{\dot{\varphi}\})$.
(3) $\text{valid}(G_{\text{out}}, \{\varphi\})$ *if and only if* $\text{valid}(G_{\text{ext}}, \{\ddot{\varphi}\})$.

## 6 CHECKING WHETHER ISOUTPUTSHAPE

Algorithm 2 (IsOUTPUTSHAPE) checks for a given shape $s$ if the rewritten shape $\ddot{s}$ is entailed by a set of axioms valid for every

(a) $G_{\text{in}}$  (b) $\dot{G}_{\text{med}}$  (c) $G_{\text{V}}$  (d) $\ddot{G}_{\text{out}}$

**Figure 2: On the left the graph $G_{\text{ext}}$ as the union of $G_{\text{in}}$ (a), $\dot{G}_{\text{med}}$ (b), $G_{\text{V}}$ (c), and $\ddot{G}_{\text{out}}$ (d).**

extended graph $G_{\text{ext}}$ derived from $q$ and $\mathcal{S}_{\text{in}}$. Based on Proposition 3, Corollary 1 establishes the formal foundation for IsOutputShape.

**Corollary 1.** *Let $q$ be a SCCQ, $\Sigma$ a set of $\mathcal{ALCHOI}$ axioms such that* valid$(G_{\text{ext}}, \Sigma)$ *for every extended graph $G_{\text{ext}}$ of $q$, and $s$ a shape including no names with dots. If $\Sigma \models \ddot{s}$, then* valid$(G_{\text{out}}, \{s\})$ *for every output graph $G_{\text{out}}$ of $q$ .*

In the remainder of this section, we will construct such a set of axioms $\Sigma$. We start by inferring the assumptions of the validation knowledge base of $G_{\text{ext}}$ based on the atoms of the input query. Next, we identify subsumptions between query variables in different components of the input query by establishing a mapping between them. Finally, we include subsumptions between role names by considering the query variables constraining them. In the *appendix* we prove that Problem IsOutputShape is NP-hard.

---

**Algorithm 2** IsOutputShape : $(\mathcal{S}_{\text{in}}, q, s) \mapsto \{\text{YES}, \text{NA}\}$

---

**Input** A finite set of shapes $\mathcal{S}_{\text{in}}$, a SCCQ $q = H \leftarrow P$, and a shape $s$ that is relevant for this query $q$.

**Output** Does valid$(\llbracket q \rrbracket_{G_{\text{in}}}, \{s\})$ hold for every graph $G_{\text{in}}$ where valid$(G_{\text{in}}, \mathcal{S}_{\text{in}})$?

1: $\Sigma_{\text{in}} \leftarrow \mathcal{S}_{\text{in}}$
2: $\Sigma_{\text{vkb}} \leftarrow \text{UNA}(q) \cup \text{CWA}(q)$
3: $\Sigma_{\text{map}} \leftarrow \text{MA}_{\mathcal{S}_{\text{in}}}(P)$
4: $\Sigma_{\text{prop}} \leftarrow \text{RS}(q)$
5: $\Sigma \leftarrow \Sigma_{\text{in}} \cup \Sigma_{\text{vkb}} \cup \Sigma_{\text{map}} \cup \Sigma_{\text{prop}}$
6: **return** **if** $\Sigma \models \ddot{s}$ **then** YES **else** NA

---

## 6.1 Axiomatizations from Validation KB Assumptions

We first utilize the assumptions of the validation knowledge base (see Definition 4) to infer axioms from a query $q$ that are valid on *any* extended graph of $q$. Since we do not know all individual names in the extended graphs, we limit the UNA-encoding to individual names that appear in the query (Definition 13), which are in any non-empty extended graph per definition (see Definition 12).

**Definition 13** (UNA-encoding). *The UNA-encoding of a query $q$, denoted* UNA$(q)$, *is the minimal set of $\mathcal{ALCHOI}$ axioms such that for every pair of distinct individual names $a, b$ occurring in $q$, the axiom $\{a\} \sqcap \{b\} \equiv \perp$ is in* UNA$(q)$.

**Proposition 4.** *For every extended graph $G_{\text{ext}}$ of a SCCQ $q$, it holds that* valid$(G_{\text{ext}}, \text{UNA}(q))$.

We do not infer any axioms based on the DCA because a SCCQ does not determine the set of individual names **I**. Concerning the CWA, a query imposes restrictions on concept names that appear

in the query pattern (e.g., $\dot{A}$), the query template (e.g., $\ddot{A}$), variables (e.g., $V_x$), and individual names (e.g., $\{a\}$). All other concept names are irrelevant (see Proposition 2).

We define the following utility functions $\text{C}_u$ (Definition 14) for referring to the nominal concept *or* variable concept for an individual name *or* variable $u$, and vcg$(q)$, referring to the variable connectivity graph of a query $q$.

**Definition 14.** *For each individual name or variable $u$, $\text{C}_u$ is $\{a\}$ if $u$ is an individual name $a$, or $\text{C}_u$ is $V_x$ if $u$ is a variable $x$.*

**Definition 15** (Variable Connectivity Graph). *The variable connectivity graph of query pattern $P$, denoted vcg$(P)$, is the graph whose nodes are the atoms in $P$, and which has an undirected edge $\{t_1, t_2\}$ if and only if atoms $t_1$ and $t_2$ share a variable.*

A SCCQ imposes restrictions on concept names in extended graphs, by definition of $G_{\text{med}}$, $G_{\text{out}}$, and $G_{\text{V}}$. For example, each atom $x{:}A \in P$ implies $V_x \sqsubseteq \dot{A}$, since concept $V_x$ is defined from all individual names referred to by $x$, which according to the evaluation semantics of SCCQ result from filtering $A$. More generally, all atoms $x{:}A$ and $(x, y){:}p$ in $P$ (Example 4) restrict the instances of variable concept $V_x$. These observations can be combined over all atoms in a query, leading to Definition 16.

**Definition 16** (CWA-encoding). *The CWA-encoding for a SCCQ $q = (H \leftarrow P)$, denoted CWA$(q)$, is the minimal set of $\mathcal{ALCHOI}$ axioms including:*

1. For each concept name $A$ in $P$, $\dot{A} \equiv A \sqcap \bigsqcup_{u:A \in P} \text{C}_u$.
2. For each concept name $A$ in $H$, $\ddot{A} \equiv \bigsqcup_{u:A \in H} \text{C}_u$.
3. For each variable $x$ in var$(q)$ the axiom

$$V_x \sqsubseteq \bigsqcap_{x:A \in P} A \sqcap \bigsqcap_{(x,u):p \in P} \exists p.\,\text{C}_u \sqcap \bigsqcap_{(u,x):p \in P} \exists p^-.\text{C}_u,$$

and if vcg$(P)$ is acyclic w.r.t to $x$, then also the axiom

$$V_x \sqsupseteq \bigsqcap_{x:A \in P} A \sqcap \bigsqcap_{(x,u):p \in P} \exists p.\,\text{C}_u \sqcap \bigsqcap_{(u,x):p \in P} \exists p^-.\text{C}_u\,.$$

4. For each role name $p$ in pattern $P$ the axioms

$$\exists \dot{p}.\,\text{C}_v \equiv \bigsqcup_{(u,v):p \in P} \text{C}_u, \qquad \exists \dot{p}.\top \equiv \bigsqcup_{(u,v):p \in P} \text{C}_u \sqcap \exists \dot{p}.\,\text{C}_v,$$

$$\exists \dot{p}^-.\,\text{C}_u \equiv \bigsqcup_{(u,v):p \in P} \text{C}_v, \qquad \exists \dot{p}^-.\top \equiv \bigsqcup_{(u,v):p \in P} \text{C}_v \sqcap \exists \dot{p}^-.\text{C}_u\,.$$

5. For each role name $p$ in template $H$ the axioms

$$\exists \ddot{p}.\,\text{C}_v \equiv \bigsqcup_{(u,v):p \in H} \text{C}_u, \qquad \exists \ddot{p}.\top \equiv \bigsqcup_{(u,v):p \in H} \text{C}_u \sqcap \exists \ddot{p}.\,\text{C}_v,$$

$$\exists \ddot{p}^-.\,\text{C}_u \equiv \bigsqcup_{(u,v):p \in H} \text{C}_v, \qquad \exists \ddot{p}^-.\top \equiv \bigsqcup_{(u,v):p \in P} \text{C}_v \sqcap \exists \ddot{p}^-.\text{C}_u\,.$$

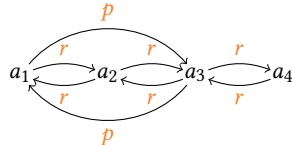

**Figure 3: Input graph $G$ for Example 6.**

Observe, that unlike in the definition for concepts $\dot{A}$ (Definition 16, 1.), the definition for concepts $\ddot{A}$ (Definition 16, 2.) does not include $A$, since elements of $\dot{A}$ are the result of filtering $A$, whereas $\ddot{A}$ is newly constructed for the query template $H$. We first demonstrate the general meaning of these axioms in Example 5.

**Example 5.** Consider again the query $q_1 = \{y{:}E, z{:}B, (y,z){:}p\} \leftarrow \{(w,y){:}p, y{:}B, (x,z){:}p, z{:}E\}$ (Example 2). Then, $\mathrm{CWA}(q_1)$ consists of the following axioms:

(1) $\{\dot{B} \equiv B \sqcap V_y, \dot{E} \equiv E \sqcap V_z\} \subset \mathrm{CWA}(q_1)$, because, e.g., concept $\dot{B}$ in the extended graph is defined by filtering $B$ with variable $V_y$ (based on the query pattern $y{:}B$ in $q_1$).

(2) $\{\ddot{B} \equiv V_z, \ddot{E} \equiv V_y\} \subset \mathrm{CWA}(q_1)$, because, e.g., concept $\ddot{B}$ in the extended graph is defined by $V_z$, since it only occurs in the single construct pattern $z{:}B$. If there were multiple occurences, it would be defined by the union of all variables, instead.

(3) $\{V_w \sqsubseteq \exists p.V_y, V_x \sqsubseteq \exists p.V_z, V_y \sqsubseteq \exists p.V_w \sqcap B, V_z \sqsubseteq \exists p.V_x \sqcap E\} \subset \mathrm{CWA}(q_1)$, because variable concepts are defined by the constraints to the respective variable in the query pattern. For example, variable concept $V_y$ is constrained by patterns $(w,y){:}p$ and $y{:}B$ in $q_1$, and thus bound by $\exists p.V_w \sqcap B$. This is a crucial step of the algorithm, since concept and role names in the extended graph are defined in terms of these variable concepts. The inverse cases are included, because $\mathrm{vcg}(P)$ is acyclic: $\{V_w \sqsupseteq \exists p.V_y, V_x \sqsupseteq \exists p.V_z, V_y \sqsupseteq \exists p.V_w \sqcap B, V_z \sqsupseteq \exists p.V_x \sqcap E\} \subset \mathrm{CWA}(q_1)$. See also Example 6 for why this condition is required.

(4) $\{\exists \dot{p}.V_y \equiv V_w, \exists \dot{p}.V_z \equiv V_x, \exists \dot{p}.\top \equiv (V_w \sqcap \exists \dot{p}.V_y) \sqcup (V_x \sqcap \exists \dot{p}.V_z)\} \subset \mathrm{CWA}(q_1)$, because, e.g., role name $\dot{p}$ in the extended graph is defined by the variables concepts that it occurs with. Similarly, the following axioms for inverse role names are included: $\{\exists \dot{p}^-.V_w \equiv V_y, \exists \dot{p}^-.V_x \equiv V_z, \exists \dot{p}^-.\top \equiv (V_y \sqcap \exists \dot{p}^-.V_w) \sqcup (V_z \sqcap \exists \dot{p}^-.V_x)\} \subset \mathrm{CWA}(q_1)$.

(5) $\{\exists \ddot{p}.V_z \equiv V_y, \exists \ddot{p}.\top \equiv V_y \sqcap \exists \ddot{p}.V_z\} \subset \mathrm{CWA}(q_1)$, because similarly to the previous case, e.g., role name $\ddot{p}$ in the extended graph is defined by the variables concepts that it occurs with. And again, similarly for inverse role names: $\{\exists \ddot{p}^-.V_y \equiv V_z, \exists \ddot{p}^-.\top \equiv V_z \sqcap \exists \ddot{p}^-.V_y\} \subset \mathrm{CWA}(q_1)$.

Note the additional condition in the second part of (Definition 16, 3.), where we require $\mathrm{vcg}(q)$ to be acyclic. In the following example (Example 6), we will motivate why this condition is required and then define Lemma 1 with respect to this case.

**Example 6.** Consider the pattern $P = \{(x,y){:}r, (y,z){:}r, (x,z){:}p\}$ of a query $q = H \leftarrow P$, and the graph $G$ in Figure 3. Note, that $\mathrm{vcg}(P)$ is cyclic, since $((x,y){:}r, (y,z){:}r)$, $((y,z){:}r, (x,z){:}p)$ as well as $((x,z){:}p, (x,y){:}r)$ each share variables.

Evaluating $q$ on $G$ results in mappings $\mu_1 = \{x \mapsto a_1, y \mapsto a_2, z \mapsto a_3\}$ and $\mu_2 = \{x \mapsto a_3, y \mapsto a_2, z \mapsto a_1\}$. Thus, the variable concepts are defined as $V_x = \{a_1, a_3\}$, $V_y = \{a_2\}$ and $V_z = \{a_3, a_1\}$. Note, that $y \mapsto a_4$ is not in any result mapping for query $q$ on graph $G$ (and $a_4 \notin V_y$). However, $\{a_4\} \sqsubseteq \exists r^-.V_x \sqcap \exists r.V_z$. Therefore, $V_y \not\sqsupseteq \exists r^-.V_x \sqcap \exists r.V_z$, so we can not include this axiom.

Intuitively, an acyclic graph $\mathrm{vcg}(P)$ allows for separating the pattern $P$ (given, as an example, variable $x$ and concept name $A$) into patterns $P_l$, $\{x : A\}$, and $P_r$, where $P_l$ shares at most variable $x$ with $P_r$. In these cases, the implicit dependencies between bindings for variables that cause issues as demonstrated for $x$ and $z$ in Example 6 do not occur.

**Lemma 1.** *Let $q = H \leftarrow P$ be a query such that $\mathrm{vcg}(P)$ is acyclic. Let $G$ be a graph, and let $x$ be a variable corring in $P$. Then*

$$V_x \sqsupseteq \textstyle\prod_{x:A\in P} A \sqcap \prod_{(x,u):p\in P} \exists p.\, \mathrm{C}_u \sqcap \prod_{(u,x):p\in P} \exists p^-.\, \mathrm{C}_u\,.$$

Given Lemma 1, the following proposition holds.

**Proposition 5.** *For every extended graph $G_{\mathrm{ext}}$ of a SCCQ $q$, it holds that* $\mathrm{valid}(G_{\mathrm{ext}}, \mathrm{CWA}(q))$, *if either $q$ does not include any individual names, or the output graph is guaranteed to be non-empty.*

Note the additional condition in Proposition 5: If the output graph is empty and the query includes individual names, then $G_{\mathrm{ext}}$ may not be valid with respect to $\mathrm{CWA}(q)$, since the constructed axioms may include individual names that are not guaranteed to exist. This could be remedied by not allowing individual names in queries; however, since Simple SHACL shapes do not allow individual names, these axioms do not impact soundness of the method.

## 6.2 Axiomatizations for Query Subpatterns

We refer to a pattern $P' \subseteq P$ as a *component* of the pattern $P$, if $\mathrm{vcg}(P')$ (see also Definition 15) is a connected subgraph of $\mathrm{vcg}(P)$ and there exists no $P''$ such that $P' \subset P''$ and $\mathrm{vcg}(P'')$ is a connected subgraph of $\mathrm{vcg}(P)$.

**Example 7.** Query $q_1$ (Example 2) has components $\{(w,y){:}p, y{:}B\}$ and $\{(x,z){:}p, z{:}E\}$. The CWA encoding (Example 5) does not entail $V_y \sqsubseteq V_z$, even though this axiom is both valid in all extended graphs, and required for inferring, e.g., the result shape $E \sqsubseteq B$.

Example 7 shows that the CWA encoding alone is not sufficient for inferring all subsumptions between variable concepts. If we could find a homomorphism between two components of the query pattern, we would know that the valuations of one component are a subset of the valuations of the other component (modulo variable names), and thus infer subsumptions between variable concepts.

**Definition 17** (Component Map). For components $P_1$ and $P_2$ of $P$, every function $h : \mathrm{var}(P_1) \to \mathbf{I} \cup \mathbf{V}$ such that $P_1^h \subseteq P_2$ is called a *component map* on $P$, where we write $P_1^h$ to mean substitution of each variable $x$ in $P_1$ by $h(x)$.

**Definition 18** (Component Map Axioms). The set of axioms inferred from a component map $h$ on $P$, denoted $\mathrm{MA}_h(P)$, is the minimal set containing axiom $\mathrm{C}_{h(x)} \sqsubseteq V_x$ for every variable $x$ in the domain of $h$. The union of all sets $\mathrm{MA}_h(P)$ of a graph pattern $P$ is called $\mathrm{MA}(P)$.

**Example 8.** Consider two components $P_1 = \{(x, y):p\}$ and $P_2 = \{(z, z):p, z:A\}$. Then we can define the mapping $h(x) = z$ and $h(y) = z$, such that $P_1^h \subseteq P_2$. Therefore, we can construct the axioms $V_z \sqsubseteq V_x$ and $V_z \sqsubseteq V_y$ valid on $G_{ext}$.

**Proposition 6.** *For every extended graph $G_{ext}$ of a SCCQ $q = H \leftarrow P$, it holds that* $\text{valid}(G_{ext}, \text{MA}(P))$.

## 6.3 Extending Query Patterns by Shape Constraints

The basic mapping $\text{MA}(P)$ is not sufficient for inferring certain crucial variable concept subsumptions, as Example 9 shows.

**Example 9.** Consider components $P_1 = \{(x, z):p, z:E\}$ and $P_2 = \{(w, y):p, y:B\}$ of query $q_1$ (Example 2). Here, we can not find a mapping $h$ satisfying Definition 17. However, we know based on $S_1$ that $B \sqsubseteq E$ (Example 1). We can utilize this knowledge to extend component $P_2$, adding the pattern $y:E$ which does not alter the queries results. Now, we can find the mapping $h(x) = w, h(z) = y$ such that $P_1^h \subseteq P_2$.

Intuitively, by extending a component as illustrated in Example 9, we reveal a subsumption relationship that was implicit in the input shapes. For the same reason, the extended component is not more restrictive than the original one. We now show how this approach can be generalized.

**Definition 19** (Target Variables). A variable $x$ is *target variable* for a shape $\psi \sqsubseteq \phi$ in an atomic pattern $t$ if and only if either

(1) $t = x:A$ and $\psi = A$,
(2) $t = (x, y):p$ and $\psi = \exists p.\top$, or
(3) $t = (y, x):p$ and $\psi = \exists p^-.\top$.

**Definition 20** (Extension). The *extension* $\text{Ext}(x, \phi)$ of a variable $x$ with respect to a shape constraint $\phi$ and component $P_i$ is the set of atoms defined below, where $x_0$ is a fresh variable.

$$\text{Ext}(x, A) = \{x:A\},$$
$$\text{Ext}(x, \exists p^-.A) = \{(x_0, x):p, x_0:A\},$$
$$\text{Ext}(x, \forall p.A) = \{y:A \mid (x, y):p \in P_i\},$$
$$\text{Ext}(x, \exists p.A) = \{(x, x_0):p, x_0:A\}, \text{ and}$$
$$\text{Ext}(x, \forall p^-.A) = \{y:A \mid (y, x):p \in P_i\}.$$

Since new atoms are added to the pattern, they can be targets of input shapes, too. The recursive extension is bound by the maximum degree and diameter of the connectivity graph $\text{vcg}(P)$ of the query pattern $P$ (Definition 15).

**Definition 21** (Bound extension). Let $P_i$ be a component of a query pattern $P$, $x$ be a variable in $\text{var}(P_i)$, $S_{in}$ be a finite set of shapes, and $P_i^x$ be a pattern that results from adding iteratively atoms $\text{Ext}(u, s)$ to $P_i$, where $s \in S_{in}$, $u$ is a target variable for $s$, and either $u = x$ or $u \notin \text{var}(P_i)$. Then, $P_i^x$ is a *bound $x$-extension* of $P_i$ using $S_{in}$ if and only if the followings conditions are satisfied:

(1) the maximum degree of $\text{vcg}(P_i^x)$ is not bigger than the maximum degree of $\text{vcg}(P)$,
(2) the diameter of graph $\text{vcg}(P_i^x \setminus P_i)$ is not longer than the maximum diameter of the components of $\text{vcg}(P)$.

**Definition 22** (Maximum Extension). Given a component $P_i$ of pattern $P$, a variable $x \in \text{var}(P_i)$, and a finite set of Simple SHACL shapes $S_{in}$, $\text{MaxExt}_x(P_i, S_{in})$ is the maximum bound $x$-extension for $P_i$ using $S_{in}$. The *maximum extension* for $P_i$ using $S_{in}$, denoted $\text{MaxExt}(P_i, S_{in})$, is the pattern $\bigcup_{x \in \text{var}(P_i)} \text{MaxExt}_x(P_i, S_{in})$.

Intuitively, Definition 21 and Definition 22 ensure that an extended component is finite, but still allows for all possible mappings with another component: Since we are only interested in finding axioms involving names in $G_{ext}$, we must use at least one such name in the mapping. Since the other mapping component is a subset of $P$, the mapping can then, in the worst case, only extend with respect to the maximum degree and diameter of $P$.

The maximum extension thus allows for finding all axioms of interest via component maps $h$ from $P_1$ to $\text{MaxExt}(P_2, S_{in})$, where $P_1$ and $P_2$ are components of $P$.

**Definition 23** (Extended Component Map Axioms). The set of *extended component map axioms* of a pattern $P$, and a set of shapes $S_{in}$, denoted $\text{MA}_{S_{in}}(P)$ is the set that includes an axiom $C_u \sqsubseteq V_x$ if and only if there is a pair of components $P_1$ and $P_2$ of $P$, and a component map $h$ from $P_1$ to $\text{MaxExt}(P_2, S_{in})$ such that $h(x) = u$ and $u$ is a variable or an individual name occurring in $P_2$.

**Proposition 7.** *For every extended graph $G_{ext}$ of a SCCQ $q = H \leftarrow P$ and set of input shapes $S_{in}$, it holds that* $\text{valid}(G_{ext}, \text{MA}_{S_{in}}(P))$.

## 6.4 Axiomatizations for Role Hierachies

Not only variable concepts form hierarchies that are not entailed by the axioms included this far. We finally infer axioms representing additional role hierarchies, that are determined from the query (Definition 24).

**Definition 24** (Role Hierarchy Axioms). The *role hierarchy axioms* of a query $q = (H \leftarrow P)$ are the set of axioms, denoted $\text{RS}(q)$, that include:

(1) for each role name $p \in P$, the axiom $\dot{p} \sqsubseteq p$,
(2) for each role name $p \in P$, the axiom $p \sqsubseteq \dot{p}$, if all atoms with role name $p$ occurring in $P$ have the form $(x, y):p$ where variables $x$ and $y$ occur in no other atom in $P$ and $x \neq y$,
(3) for each pair of role names $p, r$ with $(x, y):p \in P$ and either $(x, y):r \in H$ or $(y, x):r \in H$
 (a) the axiom $\dot{p} \sqsubseteq \ddot{r}$ (if $(x, y):r \in H$) or the axiom $\dot{p} \sqsubseteq \ddot{r}^-$ (if $(y, x):r \in H$), if $P$ does not contain any other atoms with role name $p$, and
 (b) the axiom $\ddot{r} \sqsubseteq \dot{p}$ (if $(x, y):r \in H$) or the axiom $\ddot{r}^- \sqsubseteq \dot{p}$ (if $(y, x):r \in H$), if $H$ does not contain any other atoms with role name $r$.

Trivially, for any role name $p \in P$, the axiom $\dot{p} \sqsubseteq p$ holds, since, by definition, $G_{med} \subseteq G_{in}$. The inverse axiom $p \sqsubseteq \dot{p}$ holds, if the role name is unconstrained in pattern $P$. Role hierarchy axioms between $p \in P$ and $r \in H$, that is axioms $\ddot{r} \sqsubseteq \dot{p}$ and $\dot{p} \sqsubseteq \ddot{r}$, hold, if there are no further restrictions on $r$ and $p$, respectively.

**Example 10.** Consider the input shape $A \sqsubseteq \exists p.A$ and the query $q_2 = \{(x, y):p, z:A\} \leftarrow \{(x, y):p, z:A\}$. The axioms presented prior to Definition 24 do not entail the shape $\ddot{A} \sqsubseteq \exists \ddot{p}.\ddot{A}$, even though $A \sqsubseteq \exists p.A$ should apply to the output graph: After all, we simply copy all instances of $A$ and the entirety of $p$. If we include, however,

axioms $p \sqsubseteq \dot{p}$ and $\dot{p} \sqsubseteq p$, the new set of axioms does indeed entail $\ddot{A} \sqsubseteq \exists \ddot{p}.\ddot{A}$, as expected. We can include these axioms – in this case – since we simply copy $p$ in its entirety, or, formally, the variables in the only atomic pattern including the role name $p$ are not further constrained.

**Proposition 8.** *For every extended graph* $G_{\text{ext}}$ *of a SCCQ* $q$, *it holds that* $\text{valid}(G_{\text{ext}}, \text{RS}(q))$.

## 7 IMPLEMENTATION

We implemented Algorithm 1, relying on a straightforward translation of Algorithm 2 to Scala for validation with respect to a single candidate shape, and a generator for candidates based on the syntax of Simple SHACL. For reasoning tasks, our implementation supports any OWL API reasoner. In particular, we rely on HermiT [3]. The implementation also features tools for generative exploration regarding query and vocabulary size, a test suite, and the examples from this paper in mechanized form.

We showed feasibility in experiments. To this end, we defined three sample configurations SMALL (1-2 atomic patterns in template and pattern, and 1-2 shapes), MEDIUM (5-7 each) and LARGE (11-13 each) for generating random queries based on real-world query dimensions. Shapes are generated from the vocabulary of the query. Thus, the number of input shapes given here is not comparable to the size of usual sets of SHACL shapes in real-world datasets, but rather constrain the query very tightly. We obtained the following results, running 5.000 samples (after warmup):

- SMALL: Average 3ms, median 0ms
- MEDIUM: Average 40ms, median 20ms
- LARGE: Average 693ms, median 243ms

We refer to the *appendix* for details on our methodology, reproducibility, and full results.

The implementation will be published under a free software license upon publication of this paper. An anonymized version is made available to reviewers for download here[1].

## 8 RELATED WORK

The problem of automatically inferring SHACL (or ShEx [21]) shapes from various inputs has been studied before. Most commonly it has been considered in the context of constructing shapes from concrete instance data, based on summaries of statistical information over graphs [12, 22, 25], or more involved machine learning techniques [13, 18, 19]. Some approaches combine such methods with tools for manual exploration and adaptation of inferred schemata [7]. Our approach, on the other hand, allows the construction of valid shapes from only input shapes and a given query, without the need to consider (or indeed provide) any concrete instance data.

Our work is based in the correspondence of SHACL and description logics, inspired by Bogaerts et al. [6]. This correspondence has been investigated before. Astrea [10] produces SHACL shapes from OWL ontologies by providing a mapping relating patterns of ontology constructs (i.e., language constructs including a specific usage context) with equivalent patterns of SHACL constructs validating

them. Similarly, Pandit et al. [20] explore the usage of ontology design patterns for the generation of SHACL shapes.

Inference of constraints, as well as SHACL shapes, from other data formalisms has been studied before. Calvanese et al. [9] and Sequeda et al. [24] consider inference of RDFS and OWL, respectively, from direct mappings [2] between relational data and RDF. Similarly, Thapa and Giese [26] consider inferences of SHACL shapes from direct mappings, while RML2SHACL [11] allows the translation of RML rules to SHACL shapes. These approaches differ from our approach, in that the input is restricted to a direct mapping or RML mapping from relational data, whereas in our case, the input is defined by an arbitrary query pattern imposing additional constraints, as well as constraints explicitly given as input shapes.

Finally, Thapa and Giese [27] also consider mappings of relational data to RDF. In particular, the work focuses on including SQL integrity constraints (keys, uniqueness and not-null constraints) in the translation to SHACL constraints, allowing for a limited number of property constraints mapped from integrity constraints.

## 9 CONCLUDING REMARKS

We have presented an algorithm for constructing a set of shapes characterizing the possible output graphs of CONSTRUCT queries, where the input graphs of these queries can be constrained by a set of shapes as well. The shapes are expressed in a subset of SHACL, whereas the queries are expressed in a subset of SPARQL. This enables the inference of shapes over result graphs of data processing pipelines (i.e., compositions of CONSTRUCT queries), which can be used both for validation purposes when working with these result graphs, and informatively, aiding developers directly.

The algorithm decides for the finite set of candidate shapes, whether they are entailed by a set of description-logic axioms valid on the union of graphs involved in the query operation. We proof soundness of this algorithm, and provide an implementation.

*Limitations.* (1) The set of output shapes computed by our approach is sound, but incomplete. Consider as an example the problem $q_3 = \{(x, y){:}p, z{:}A\} \leftarrow \{(x, y){:}p, z{:}A, (z, w){:}p\}$ and $\mathcal{S}_{\text{in}} = \{A \sqsubseteq \exists p.A\}$. Here, the input shape would apply to the output, but we can not infer it. This and similar problems could perhaps be remedied by extending the inference of role hierachy axioms to also consider input shapes.

(2) Our approach is limited to a subset of SHACL and SPARQL. In the *appendix*, we show how to extend the approach to arbitrary $\mathcal{ALCHOI}$ constraints. Intuitively, this extension is possible because the propositions presented in this paper are not restricted to Simple SHACL (consider, in particular, Proposition 3).

*Future Work.* In order to extend the approach to queries involving generic patterns (e.g., $(x, y){:}z$ or $x{:}z$), an expansion to non-generic queries may be possible, since all relevant role and concept names are known from template and input shapes.

While the application of our approach to entire data processing pipelines is straightforward, there are interesting empirical questions regarding the properties of results shapes, e.g., depending on the nature of input shapes or number of processing steps left as future work.

---

[1]https://e.pcloud.link/publink/show?code=XZj6rWZAxwMxsCg2jfQO2ptF96HUb72o5bk

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

# A  STRUCTURE OF THE APPENDIX

The appendix is structured as follows. In Appendix B we give an extended version of the running examples used throughout the paper. Appendix C contains the full proofs for all propositions from the main paper. Appendix D details how the method from the main paper can be extended for more general types of SHACL shapes, and Appendix E gives proofs related to these extensions. Finally, Appendix F contains some details about the implementation.

# B  EXTENDED EXAMPLES

In this section, we extend upon the examples from the main paper. We first provide additional details on the running example incorporated in the body of the paper, including the example using concrete SHACL and SPARQL syntax. Next, we extend upon the running example by giving additional example queries, and more examples beyond that.

## B.1  Implementation

All examples from the main paper and from this section are also provided in mechanized form with the implementation. To this end, the implementation contains .shacl and .sparql files, where example shapes are included in formal description logics syntax, and SPARQL queries in concrete SPARQL syntax. We refer to the documentation (Readme.md) for more details on running example instances, and obtaining different kinds of outputs.

## B.2  Running Example: Full Set of Shapes $\mathcal{S}_{out}$

The full set of output shapes from Example 3 is given below. Note, that some shapes (such as tautologies and shapes trivially entailed by other shapes) are omitted.

**Example 11.** Full output shapes for $q_1 = \{y{:}E, z{:}B, (y, z){:}p\} \leftarrow \{(w, y){:}p, y{:}B, (x, z){:}p, z{:}E\}$ (Example 2) and the set of input shapes $S_1 = \{A \sqsubseteq \exists p.B, \exists r.\top \sqsubseteq B, B \sqsubseteq E\}$ (Example 1), as first introduced

```
:s1  a  sh:NodeShape  ;
     sh:targetClass  :A  ;
     sh:property  [
          sh:path  :p  ;
          sh:class  :B  ;
          sh:minCount  1  ;
     ]  .

:s2  a  sh:NodeShape  ;
     sh:targetSubjectsOf  :r  ;
     sh:class  :B  .

:s3  a  sh:NodeShape  ;
     sh:targetClass  :B  ;
     sh:class  :E  .
```

**Figure 4: Shapes** $s_1$ ($A \sqsubseteq \exists p.B$), $s_2$ ($\exists r.\top \sqsubseteq B$), **and** $s_3$ ($B \sqsubseteq E$) **using a concrete SHACL syntax (Turtle).**

in Example 3.

$S_{1\text{-out}} = \{$

| | | |
|---|---|---|
| $B \sqsubseteq \forall p^-.B,$ | $B \sqsubseteq \forall p^-.E,$ | $B \sqsubseteq \forall p.B,$ |
| $B \sqsubseteq \exists p^-.B,$ | $B \sqsubseteq \exists p^-.E,$ | $E \sqsubseteq B,$ |
| $E \sqsubseteq \forall p^-.B,$ | $E \sqsubseteq \forall p^-.E,$ | $E \sqsubseteq \forall p.B,$ |
| $E \sqsubseteq \exists p^-.B,$ | $E \sqsubseteq \exists p^-.E,$ | $E \sqsubseteq \exists p.B,$ |
| $\exists p^-.\top \sqsubseteq B,$ | $\exists p^-.\top \sqsubseteq \forall p^-.B,$ | $\exists p^-.\top \sqsubseteq \forall p^-.E,$ |
| $\exists p^-.\top \sqsubseteq \forall p.B,$ | $\exists p^-.\top \sqsubseteq \exists p^-.B,$ | $\exists p^-.\top \sqsubseteq \exists p^-.E,$ |
| $\exists p.\top \sqsubseteq B,$ | $\exists p.\top \sqsubseteq E,$ | $\exists p.\top \sqsubseteq \forall p^-.B,$ |
| $\exists p.\top \sqsubseteq \forall p^-.E,$ | $\exists p.\top \sqsubseteq \forall p.B,$ | $\exists p.\top \sqsubseteq \exists p^-.B,$ |
| $\exists p.\top \sqsubseteq \exists p^-.E,$ | $\exists p.\top \sqsubseteq \exists p.B \}$ | |

## B.3 Running Example: Concrete Syntax

We next give the running example (e.g., Example 11) in concrete SPARQL and SHACL (Turtle) syntax. We assume the default prefix : for the example domain (unspecified), and prefix sh: for SHACL (i.e., bound to http://www.w3.org/ns/shacl#).

## B.4 Additional Examples

We now give additional examples problem instances, that is, queries and sets of input shapes, and (a subset of) the corresponding output shapes. For the full output, as well as all intermediate components, i.e., the inferred axioms, we refer to the implementation, which renders full internal details via the −debug flag. All examples are included with the implementation.

**Example 12.** With input $q_4 = \{x{:}B, y{:}A, \} \leftarrow \{x{:}A, y{:}B, \}$ and $S_4 = \{A \sqsubseteq B\}$, we obtain the set $\{B \sqsubseteq A\}$ as output.

Query $q_4$ is a simple example, demonstrating how our method maintains subsumption relationships through renaming of concepts, in this simple case swapping of $A$ and $B$. A core component of this is the subsumption between the variable concept for query variables

```
CONSTRUCT  {
     ?y  a  :E  .
     ?z  a  :B  .
     ?y  :p  ?z
}  WHERE  {
     ?w  :p  ?y  .
     ?y  a  :B  .
     ?x  :p  ?z  .
     ?z  a  :E
}
```

**Figure 5: Example query** $q_1 = \{y{:}E, z{:}B, (y, z){:}p\} \leftarrow \{(w, y){:}p, y{:}B, (x, z){:}p, z{:}E\}$ **in concrete SPARQL syntax.**

?x and ?y in the query, which holds on all extended graphs for $q_4$ and $S_4$.

**Example 13.** With input $q_5 = \{x{:}B, y{:}A, \} \leftarrow \{x{:}A, (x, y){:}p, y{:}B, \}$ and $S_5 = \{B \sqsubseteq A, B \sqsubseteq \exists p.B\}$, we obtain the set $\{A \sqsubseteq B\}$ as output.

Here, we continue with another example with the same, simple template as in the previous example. This simple template serves to demonstrate the consequences of variable concept subsumption between the variables ?y and ?x directly, as the output shape $A \sqsubseteq B$. This subsumption relationship results from the mapping step discussed in Section 6: Since we know for all bindings of $y$ in the query, that both the pattern $y{:}A$ is always satisfied (since $B \sqsubseteq A$) and the same for $(y, z){:}p, z{:}B$ (for some fresh variable $z$, since $B \sqsubseteq \exists p.B$), we can obtain a mapping resulting in subsumption $V_y \sqsubseteq V_x$.

**Example 14.** With input $q_6 = \{x{:}A, y{:}B, (x, y){:}p\} \leftarrow \{x{:}A, y{:}B, \}$ and $S_6 = \{\}$, we obtain the set $\{A \sqsubseteq \forall p^-.A, A \sqsubseteq \forall p.B, A \sqsubseteq \exists p.B, B \sqsubseteq \forall p^-.A, B \sqsubseteq \forall p.B, B \sqsubseteq \exists p^-.A, \exists p^-.\top \sqsubseteq B, \exists p^-.\top \sqsubseteq \forall p^-.A, \exists p^-.\top \sqsubseteq \forall p.B, \exists p^-.\top \sqsubseteq \exists p^-.A, \exists p.\top \sqsubseteq A, \exists p.\top \sqsubseteq \forall p^-.A, \exists p.\top \sqsubseteq \forall p.B, \exists p.\top \sqsubseteq \exists p.B, \}$ as output.

With this example, we demonstrate inference of shapes from the query (template) itself, without any given input shapes, resulting only from the closure assumptions used in the method. The query pattern simply introduces the variables ?x and ?y without any further context (arbitrary names $A$ and $B$). In the template, we introduce the additional role name $p$ between these two variables.

## C PROOFS

In this section, we present the full proofs for Proposition 1 through Proposition 8, and introduce Theorem 1 (and its proof) as well as Proposition 9 (and its proof).

## C.1 Proof for Proposition 1

In order to prove Proposition 1 we need to show that every model of the validation knowledge base of a graph is isomorphic to the canonical model of the graph. To this end, we introduce the following lemma.

**Lemma 2.** *Let $G$ be a graph, $\mathcal{I}_G$ the canonical interpretation of $G$, and $(\mathcal{T}_G, G)$ the validation knowledge base of $G$. Then, all models $\mathcal{I}$ of $(\mathcal{T}_G, G)$ are isomorphic to $\mathcal{I}_G$.*

PROOF. The fact that $\mathcal{I}$ and $\mathcal{I}_G$ are isomorphic follows from the existence of a function $f : \Delta^{\mathcal{I}_G} \to \Delta^{\mathcal{I}}$ satisfying the following properties:

**P.1** Function $f$ is bijective.
**P.2** $f(a^{\mathcal{I}_G}) = a^{\mathcal{I}}$ for every $a \in \mathbf{I}$.
**P.3** $\{f(x) \mid x \in A^{\mathcal{I}_G}\} = A^{\mathcal{I}}$ for every $A \in \mathbf{C}$.
**P.4** $\{(f(x), f(y)) \mid (x, y) \in r^{\mathcal{I}_G}\} = r^{\mathcal{I}}$ for every $r \in \mathbf{R}$.

We next prove properties **P.1** to **P.4** for function $f$.

**Proof for P.1:** Let $f : \Delta^{\mathcal{I}_G} \to \Delta^{\mathcal{I}}$ be the function defined as $f(a) = a^{\mathcal{I}}$, for every individual name $a \in \mathbf{I}$. Function $f$ is well-defined because, by definition of $\mathcal{I}_G$, $\Delta^{\mathcal{I}_G} = \mathbf{I}$. To show that function $f$ is bijective, it suffices to prove that $f$ is injective and surjective:

    **Surjective:** The domain closure assumption axioms in the knowledge base $(\mathcal{T}_G, G)$ imply that $\Delta^{\mathcal{I}} = \bigcup_{a \in \mathbf{I}} \{a^{\mathcal{I}}\}$. Then, for every element $e \in \Delta^{\mathcal{I}}$, there exists an individual name $\{a\}$ such that $e \in \{a\}^{\mathcal{I}}$. That is, $f(a) = e$. Hence, $f$ is surjective.

    **Injective:** The unique-name assumption axioms in the knowledge base $(\mathcal{T}_G, G)$ imply that $\mathcal{I} \models \{b\} \sqcap \{a\} \equiv \bot$ for every pair of distinct individual names $a$ and $b$. That is, $f(a) \neq f(b)$. Hence, $f$ is injective.

**Proof for P.2:** Let $a \in \mathbf{I}$ be an arbitrary individual name. By definition of $\mathcal{I}_G$, $a^{\mathcal{I}_G} = a$. By definition of $f$, $f(a) = a^{\mathcal{I}}$. Hence, combining both identities, we obtain the identity $f(a^{\mathcal{I}_G}) = a^{\mathcal{I}}$.

**Proof for P.3:** Let $A \in \mathbf{C}$ be an arbitrary concept name. By definition of $\mathcal{I}_G$, $A^{\mathcal{I}_G} = \{a \mid a{:}A \in G\}$. The closed-world assumption axioms in the knowledge base $(\mathcal{T}_G, G)$ imply that $A^{\mathcal{I}} = \bigcup_{a:A \in G} \{a^{\mathcal{I}}\}$. That is, $A^{\mathcal{I}} = \{a^{\mathcal{I}} \mid a{:}A \in G\}$. Since $f(a) = a^{\mathcal{I}}$, we conclude that $\{f(a) \mid a \in A^{\mathcal{I}_G}\} = A^{\mathcal{I}}$.

**Proof for P.4:** Let $r \in \mathbf{R}$ be an arbitrary role name. By definition of $\mathcal{I}_G$, $r^{\mathcal{I}_G} = \{(a, b) \mid (a, b){:}r \in G\}$. The closed-world assumption axioms in the knowledge base $(\mathcal{T}_G, G)$ imply that $(\exists r.\{b\})^{\mathcal{I}} = \bigcup_{(a,b):r \in G} \{a^{\mathcal{I}}\}$. That is, $r^{\mathcal{I}} = \{(a^{\mathcal{I}}, b^{\mathcal{I}}) \mid (a, b){:}r \in G\}\}$. Since $f(a) = a^{\mathcal{I}}$ and $f(b) = b^{\mathcal{I}}$, we conclude that $\{(f(a), f(b)) \mid (a, b) \in r^{\mathcal{I}_G}\} = r^{\mathcal{I}}$.

Hence, we have proved the lemma. □

PROOF OF PROPOSITION 1. This proof follows from Lemma 2, which states that $(\mathcal{T}_G, G)$ has a unique model up to isomorphism, namely $\mathcal{I}_G$; thus for every set $\Sigma$ of $\mathcal{ALCHOI}$ axioms, $\mathcal{I}_G \models \Sigma$ if and only if $\mathcal{I}_G \models (\mathcal{T}_G \cup \Sigma, G)$. That is, $\mathcal{I}_G \models S$ if and only if $(\mathcal{T}_G \cup S, G)$ is consistent. Hence, statements (i) and (ii) are equivalent. Similarly, statements (ii) are (iii) are equivalent because $(\mathcal{T}_G, G)$ has a unique model up to isomorphism. In general, given two sets of axioms $\Sigma_1$ and $\Sigma_2$, the consistence of $(\Sigma_1, G)$ and $(\Sigma_2, G)$ does not imply the consistency of $(\Sigma_1 \cup \Sigma_2, G)$ because the sets models of $(\Sigma_1, G)$ and $(\Sigma_2, G)$ can be non-empty and disjoint. However, in this case the implication is true because $(\mathcal{T}_G, G)$ admits a single model up to isomorphism. □

## C.2 Proof for Proposition 2

In order to prove Proposition 2, we show by contraposition that if a given shape $s$ does not satisfy the conditions of the proposition, then it is irrelevant.

To this end, we introduce two lemmas, relating the structure of a concept expression $C$ with the vocabulary of $C$ and of a given graph $G$. First, we consider concept names and existential quantification.

**Lemma 3.** *Let $C$ be a concept description defined as follows:*

$$C ::= A \mid \exists p.\top \mid \exists p^-.\top \mid \exists p.A \mid \exists p^-.A \,,$$

*where $A$ is a concept name, and $p$ is a role name. Let $G$ be a Simple RDF graph, and $(\mathcal{T}_G, G)$ be the validation knowledge base of $G$. Then, $\text{voc}(C) \not\subseteq \text{voc}(G)$ implies $(\mathcal{T}_G, G) \models C \equiv \bot$.*

PROOF. Let $\mathcal{I}$ be a model of $(\mathcal{T}_G, G)$. We will prove this lemma by proving the contraposition: the existence of an individual $a^{\mathcal{I}} \in C^{\mathcal{I}}$ implies that $\text{voc}(C) \subseteq \text{voc}(G)$. By Lemma 2, every model $\mathcal{I}$ of $(\mathcal{T}_G, G)$ is isomorphic to the canonical model of $G$. The proof follows case by case:

(1) If $C$ is $A$ then, $a{:}A \in G$. Hence, $\text{voc}(C) \subseteq \text{voc}(G)$.
(2) If $C$ is $\exists p.\top$ or $\exists p^-.\top$, then there is an individual name $b$ such that $(a, b){:}p \in G$ or $(b, a){:}p \in G$. Hence, $\text{voc}(C) \subseteq \text{voc}(G)$.
(3) If $C$ is $\exists p.A$ or $\exists p^-.A$, then there is an individual name $b$ such that $b{:}A \in G$, and $(a, b){:}p \in G$ or $(b, a){:}p \in G$. Hence, $\text{voc}(C) \subseteq \text{voc}(G)$.

Hence, we prove the lemma by contraposition. □

Next, we consider universal quantification.

**Lemma 4.** *Let $C$ be a concept description defined as follows:*

$$C ::= \forall p.A \mid \forall p^-.A \,,$$

*where $A$ is a concept name, and $p$ is a role name. Let $G$ be a Simple RDF graph, and $(\mathcal{T}_G, G)$ be the validation knowledge base of $G$. Then the following holds.*

(1) *If $p \notin \text{voc}(G)$ then $(\mathcal{T}_G, G) \models C \equiv \top$.*
(2) *If $A \notin \text{voc}(G)$ and $C$ is $\forall p.A$ then $(\mathcal{T}_G, G) \models C \equiv \neg(\exists p.\top)$.*
(3) *If $A \notin \text{voc}(G)$ and $C$ is $\forall p^-.A$ then $(\mathcal{T}_G, G) \models C \equiv \neg(\exists p^-.\top)$.*

PROOF. Let $\mathcal{I}$ be a model of the validation knowledge base of graph $G$. We prove this lemma using the equivalencies $\forall p.A \equiv \neg\exists p.\neg A$ and $\forall p^-.A \equiv \neg\exists p^-.\neg A$.

(1) If $p \notin \text{voc}(G)$ then $p^{\mathcal{I}}$ is empty, since $\mathcal{I}$ is isomorphic to the canonical interpretation of $G$ (Lemma 2). Thus, every element in the domain $\Delta^{\mathcal{I}}$ belongs to concepts $\neg\exists p.\neg A$ and $\neg\exists p^-.\neg A$. Hence, $\mathcal{I} \models C \equiv \top$.
(2) If $A \notin \text{voc}(G)$ then $A^{\mathcal{I}}$ is empty, since $\mathcal{I}$ is isomorphic to the canonical interpretation of $G$ (Lemma 2). Then, $\mathcal{I} \models \neg A \equiv \top$. Hence,
  (a) If $C$ is $\forall p.A$, then $\mathcal{I} \models C \equiv \neg(\exists p.\top)$.
  (b) If $C$ is $\forall p^-.A$, then $\mathcal{I} \models C \equiv \neg(\exists p^-.\top)$.

Since $\mathcal{I}$ is an arbitrary model of $(\mathcal{T}_G, G)$, we conclude this proof. □

Proof of Proposition 2. Let $\psi \sqsubseteq \phi$ be a Simple SHACL shape, $q$ be a SCCQ, and $G$ be a Simple RDF graph with $\text{voc}(G) \subseteq \text{voc}(q)$, and $(\mathcal{T}_G, G)$ be the validation knowledge base of graph $G$. We have the following disjoint cases:

(1) Case $\text{voc}(\psi) \not\subseteq \text{voc}(q)$. Then, by Lemma 3, $(\mathcal{T}_G, G) \models \psi \equiv \bot$ (since $\psi$ is, per definition, restricted to one of the cases covered in the lemma). Hence, shape $\psi \sqsubseteq \phi$ is not relevant (Definition 11).

(2) Case $\text{voc}(\psi) \subseteq \text{voc}(q)$ and $\phi$ has the form $\forall p.A$ or $\forall p^-.A$. We have the following subcases:
   (a) Case $p \notin \text{voc}(q)$. Then, by Lemma 4, $(\mathcal{T}_G, G) \models \phi \equiv \top$. Hence, shape $\psi \sqsubseteq \phi$ is not relevant.
   (b) Case $p \in \text{voc}(q)$ and $A \notin \text{voc}(G)$. Then, by Lemma 3, $(\mathcal{T}_G, G) \models \neg A \equiv \top$. We have the following subcases:
      (i) Case $\phi$ is $\forall p.A$, then $(\mathcal{T}_G, G) \models \phi \equiv \neg(\exists p.\top)$.
      (ii) Case $\phi$ is $\forall p^-.A$, then $(\mathcal{T}_G, G) \models \phi \equiv \neg(\exists p^-.\top)$.

(3) Case $\text{voc}(\psi) \subseteq \text{voc}(q)$ and $\text{voc}(\phi) \not\subseteq \text{voc}(q)$ and $\phi$ has not the form $\forall p.A$ or $\forall p^-.A$. Then, $\phi$ has one of the forms covered in Lemma 3, and by this lemma, $(\mathcal{T}_G, G) \models \phi \equiv \bot$. Therefore, the shape is not relevant.

(4) Case $\text{voc}(\psi) \subseteq \text{voc}(q)$ and $\text{voc}(\phi) \subseteq \text{voc}(q)$. We have the following subcases:
   (a) Shape $\psi \sqsubseteq \phi$ has the form $A \sqsubseteq A$. Then, the shape is not relevant because it is a tautology.
   (b) Shape $\psi \sqsubseteq \phi$ has not the form $A \sqsubseteq A$.

Hence, we have shown that in all cases, except for those mentioned in Proposition 2 (2.b.i, 2.b.ii, and 4.b), the shapes are not relevant. Hence, for all relevant shapes, the properties in Proposition 2 hold. □

## C.3 Proof for Proposition 3

Proof. We prove first the second case of Proposition 3, namely $\text{valid}(G_{\text{med}}, \{\varphi\})$ if and only if $\text{valid}(G_{\text{ext}}, \{\dot{\phi}\})$. The proofs for the other two cases work exactly analogously, since all three subgraphs $G_{\text{in}}$, $G_{\text{med}}$ and $G_{\text{out}}$ form distinct namespaces.

Let $\mathcal{I}_{\text{ext}}$ and $\mathcal{I}_{\text{med}}$ be the canonical models of $G_{\text{med}}$ and $G_{\text{ext}}$, respectively. To prove this case, it suffices to show that for every axiom $\varphi$ not including any names with dots (e.g., $\dot{A}$, $\ddot{A}$, $\dot{p}$, or $\ddot{p}$), and every concept expression $C$ occurring in $\varphi$, $C^{\mathcal{I}_{\text{ext}}} = C^{\mathcal{I}_{\text{med}}}$. Indeed, if this is true for every arbitrary concept expression $C$ in $\varphi$, then for every such axiom $\varphi$, $\text{valid}(G_{\text{med}}, \{\varphi\})$ if and only $\text{valid}(G_{\text{ext}}, \{\dot{\phi}\})$. By construction of $G_{\text{ext}}$, for every concept assertion $a{:}A \in G_{\text{med}}$, $a{:}\dot{A} \in G_{\text{ext}}$ if and only if $a{:}\dot{A} \in \dot{G}_{\text{med}}$, and for every role assertion $(a, b){:}p \in G_{\text{med}}$, $(a, b){:}\dot{p} \in G_{\text{ext}}$ if and only if $(a, b){:}\dot{p} \in \dot{G}_{\text{med}}$. That is, for every concept name $A \in \mathbf{C}$ and role name $p \in \mathbf{R}$, it holds that $A^{\mathcal{I}_{\text{med}}} = \dot{A}^{\mathcal{I}_{\text{ext}}}$ and $p^{\mathcal{I}_{\text{med}}} = \dot{p}^{\mathcal{I}_{\text{ext}}}$. Hence, for every concept description $C$, $C^{\mathcal{I}_{\text{med}}} = \dot{C}^{\mathcal{I}_{\text{ext}}}$.

We give below, for the sake of completeness, the remaining two cases, case 1. and case 3.

- Case 1. By construction of $G_{\text{ext}}$, for every concept assertion $a{:}A \in G_{\text{in}}$, $a{:}A \in G_{\text{ext}}$ if and only if $a{:}A \in G_{\text{in}}$, and for every role assertion $(a, b){:}p \in G_{\text{in}}$, $(a, b){:}p \in G_{\text{ext}}$ if and only if $(a, b){:}p \in G_{\text{in}}$. That is, for every concept name $A \in \mathbf{C}$ and role name $p \in \mathbf{R}$, it holds that $A^{\mathcal{I}_{\text{in}}} = A^{\mathcal{I}_{\text{ext}}}$ and $p^{\mathcal{I}_{\text{in}}} = p^{\mathcal{I}_{\text{ext}}}$. Hence, for every concept description $C$, $C^{\mathcal{I}_{\text{in}}} = C^{\mathcal{I}_{\text{ext}}}$.

- Case 3. By construction of $G_{\text{ext}}$, for every concept assertion $a{:}A \in G_{\text{out}}$, $a{:}\ddot{A} \in G_{\text{ext}}$ if and only if $a{:}\ddot{A} \in \ddot{G}_{\text{out}}$, and for every role assertion $(a, b){:}p \in G_{\text{out}}$, $(a, b){:}\ddot{p} \in G_{\text{ext}}$ if and only if $(a, b){:}\ddot{p} \in \ddot{G}_{\text{out}}$. That is, for every concept name $A \in \mathbf{C}$ and role name $p \in \mathbf{R}$, it holds that $A^{\mathcal{I}_{\text{out}}} = \ddot{A}^{\mathcal{I}_{\text{ext}}}$ and $p^{\mathcal{I}_{\text{out}}} = \ddot{p}^{\mathcal{I}_{\text{ext}}}$. Hence, for every concept description $C$, $C^{\mathcal{I}_{\text{out}}} = \ddot{C}^{\mathcal{I}_{\text{ext}}}$.

□

## C.4 Proof for Corollary 1

Proof. The proof for Corollary 1 follows immediately from case 3 of Proposition 3. Let $q$ be a SCCQ, $\Sigma$ a set of $\mathcal{ALCHOI}$ axioms such that $\text{valid}(G_{\text{ext}}, \Sigma)$ for every extended graph $G_{\text{ext}}$ of $q$, and $s$ a Simple SHACL shape such that $\Sigma \models \ddot{s}$. Then also $\text{valid}(G_{\text{ext}}, \{\ddot{s}\})$, from which by case 3 of Proposition 3 follows immediately that $\text{valid}(G_{\text{out}}, s)$. □

## C.5 Proof for Proposition 4

Proof. By construction, each axiom in $\text{UNA}(q)$ is also an axiom in the validation knowledge base of the graph $G_{\text{ext}}$ (see Definition 4). □

## C.6 Proof for Proposition 5

We first prove Lemma 1.

Proof. We prove Lemma 1 by contradiction. Let $q = H \leftarrow P$ be a query such that $\text{vcg}(P)$ is acyclic. Let $G$ be a graph, and let $x$ be a variable corring in $P$. Let $C$ be a concept defined as

$$C \equiv \bigsqcap_{x:A \in P} A \sqcap \bigsqcap_{(x,u):p \in P} \exists p. C_u \sqcap \bigsqcap_{(u,x):p \in P} \exists p^-. C_u .$$

and assume that there is an individual name $c$ in $G$ such that $c{:}C$ is valid in $G_{\text{ext}}$, but $c{:}V_x$ is not valid in $G_{\text{ext}}$.

Without loss of generality, assume that $P$ includes a single concept assertion including the variable $x$, namely $x{:}A$, and no role assertions $(x, d){:}r$ where $d$ is an individual name. Indeed, if there are serveral atoms $x{:}A_1, \ldots, x{:}A_n$ and $(x, d_1){:}r_1, \ldots, (x, d_m){:}r_m$ we can define $A \equiv A_1 \sqcap \ldots \sqcap A_m \sqcap \exists r_1.\{d_1\} \sqcap \ldots \sqcap \exists r_m.\{d_m\}$. Without loss of generality, assume that $G_{\text{ext}}$ includes the graph

$$\{a{:}V_y, (a, c){:}r, c{:}A, (c, b){:}s, b{:}V_z\}$$

Let $\Omega$ be the set of all mappings $\mu$ such that $\mu(P) \subseteq G$.

Then, by definition there exist the mappings $\mu_1, \mu_2 \in \Omega$ such that $\mu_1(y) = a$ and $\mu_2(z) = b$, but there not exists the mapping $\mu \in \Omega$ such that $\mu(x) = c$. Then, $\mu_1(x) \neq c$ and $\mu_2(x) \neq c$.

Let $P_y$ be the part of pattern $P$ which *connects* with variables $y$ and $x$, but not $z$. Let $P_z$ be the part of pattern $P$ which *connects* with variables $z$ and $x$, but not $y$. Let

$$\mu_1^y = \mu_1\big|_{\text{var}(P_y)\setminus\{x\}}, \qquad \mu_1^z = \mu_1\big|_{\text{var}(P_z)\setminus\{x\}},$$
$$\mu_2^y = \mu_2\big|_{\text{var}(P_y)\setminus\{x\}}, \qquad \mu_2^y = \mu_2\big|_{\text{var}(P_z)\setminus\{x\}}.$$

Then, $\mu_1 = \mu_1^y \cup \{x \mapsto \mu_1(x)\} \cup \mu_1^z$ and $\mu_2 = \mu_2^y \cup \{x \mapsto \mu_2(x)\} \cup \mu_2^z$. Since $\mu_1^y$ and $\mu_2^z$ share no variables, $\mu_3 = \mu_1^y \cup \{x \mapsto c\} \cup \mu_2^z$ is a mapping.

By the definition of the semantics of SCCQ, $\mu_3 \in \Omega$. Then $c{:}V_x$ is valid in $G_{\text{ext}}$. This contradicts the initial assumptions, from which we conclude $V_x \sqsupseteq C$. □

We now continue with the proof for Proposition 5.

Proof. We prove Proposition 5 by showing the validity of this proposition for the cases 1 through 5 in Definition 16. We divide the proof in two groups: First, for cases 1, 2, and 3, and then for cases 4 and 5.

*Cases 1, 2, and 3.* For cases 1 through 3, we divide the proof in two parts each, one for either inclusion (i.e., $\sqsubseteq$ and $\sqsupseteq$). To show an inclusion $A \sqsubseteq B$ in $G_{\text{ext}}$, we will assume that there exists at least one valuation $\mu$ such that $\mu(P) \subseteq G_{\text{in}}$, and then prove that inclusion $\{a\} \sqsubseteq A$ implies inclusion $\{a\} \sqsubseteq B$ for every individual name occurring in $G_{\text{ext}}$.

$1_{\sqsubseteq}$   $\dot{A} \sqsubseteq A \sqcap \bigsqcup_{u:A \in P} C_u$. Let $a$ be an arbitrary individual name such that $\{a\} \sqsubseteq \dot{A}$ is valid in $G_{\text{ext}}$. Then $a{:}\dot{A} \in G_{\text{ext}}$, so $a{:}A \in G_{\text{med}}$. Then $a{:}A \in G_{\text{in}}$ and there is an atom $v{:}A \in P$ where $v$ is either individual name $a$ or a variable $x$. If $v$ is $a$, then $\{a\} \sqsubseteq C_v$ is trivially valid in $G_{\text{ext}}$. Otherwise $v$ is $x$ and by construction, $a{:}V_x \in G_{\text{ext}}$. Then, $\{a\} \sqsubseteq C_v$ is valid in $G_{\text{ext}}$, so $\{a\} \sqsubseteq \bigsqcup_{u:A \in P} C_u$ is also valid in $G_{\text{ext}}$. Similarly, since $G_{\text{in}} \subseteq G_{\text{ext}}$, $a{:}A \in G_{\text{ext}}$, so $\{a\} \sqsubseteq A$ is valid in $G_{\text{ext}}$. Therefore, $\{a\} \sqsubseteq A \sqcap \bigsqcup_{u:A \in P} C_u$ is valid in $G_{\text{ext}}$.

$1_{\sqsupseteq}$   $\dot{A} \sqsupseteq A \sqcap \bigsqcup_{u:A \in P} C_u$. Let $a$ be an arbitrary individual name such that $\{a\} \sqsubseteq A \sqcap \bigsqcup_{u:A \in P} C_u$ is valid in $G_{\text{ext}}$. By construction, for every individual name $b$, if $b{:}A \in G_{\text{ext}}$ then $b{:}A \in G_{\text{in}}$. Since $a{:}A \in G_{\text{ext}}$, $a{:}A \in G_{\text{in}}$ holds. By definition, $\{a\} \sqsubseteq C_u$ is valid in $G_{\text{ext}}$ for at least one atom $u{:}A \in P$. If $u$ is an individual name, then $u$ is $a$, and $a{:}A \in P$. If $G_{\text{med}}$ is not empty, then $a{:}A \in G_{\text{med}}$. Otherwise, if $u$ is a variable $x$ then $C_u$ is the variable concept $V_x$, and $a{:}V_x \in G_{\text{ext}}$. By definition, $a$ is an instance of variable $x$, and thus $a{:}A \in G_{\text{med}}$. Hence, $a{:}A \in G_{\text{med}}$ in all possible cases (when $u$ is an individual name or when $u$ is a variable). By construction, $a{:}\dot{A} \in G_{\text{ext}}$ and therefore $\{a\} \sqsubseteq \dot{A}$.

$2_{\sqsubseteq}$   $\ddot{A} \sqsubseteq \bigsqcup_{u:A \in H} C_u$. Let $a$ be an arbitrary individual name such that $\{a\} \sqsubseteq \ddot{A}$ is valid in $G_{\text{ext}}$. By construction, $a{:}A \in G_{\text{out}}$ so there is an atom $v{:}A \in H$ such that $a{:}A$ is an instance of pattern $v{:}A$, and $v$ is either concept name $a$ or a variable $x$. If $v$ is $a$ then $\{a\} \sqsubseteq C_v$ is trivially valid in $G_{\text{ext}}$. Otherwise $v$ is $x$ and since $a$ is an instance of $x$, it holds that $a{:}V_x \in G_{\text{ext}}$, so $\{a\} \in C_v$ is valid in $G_{\text{ext}}$. Therefore, $\{a\} \sqsubseteq \bigsqcup_{u:A \in H} C_u$ is valid in $G_{\text{ext}}$.

$2_{\sqsupseteq}$   $\ddot{A} \sqsupseteq \bigsqcup_{u:A \in H} C_u$. Let $a$ be an arbitrary individual name such that $\{a\} \sqsubseteq \bigsqcup_{u:A \in H} C_u$ is valid in $G_{\text{ext}}$. Then there is at least one atom $v{:}A \in H$ such that $\{a\} \sqsubseteq C_v$ is valid in $G_{\text{ext}}$. If $v$ is $a$, then $a{:}A \in G_{\text{out}}$, and thus $a{:}\ddot{A} \in G_{\text{ext}}$. Otherwise $v$ is a variable $x$, and $a{:}V_x \in G_{\text{ext}}$. By the definition of variable concepts, $a{:}A \in G_{\text{out}}$, so $a{:}\ddot{A} \in G_{\text{ext}}$. Therefore, $\{a\} \sqsubseteq \ddot{A}$ is valid in $G_{\text{out}}$.

$3_{\sqsubseteq}$   *Variable ($\sqsubseteq$).* Let $a$ be an arbitrary individual name such that $\{a\} \sqsubseteq V_x$ is valid in $G_{\text{ext}}$. We show separately for each operand $k$ in the intersection, that $\{a\} \sqsubseteq k$, assuming that the respective component is defined, below.

(a) For $k = \bigsqcap_{x:A \in P} A$: If $\{a\} \sqsubseteq V_x$, then by definition $a$ is an instance of variable $x$ in $P$, i.e., $a \in \mu(x)$. Then for each concept name $A$ occurring in an atomic pattern of the form $x{:}A \in P$ there must be $a{:}A \in G_{\text{in}}$ (since

otherwise $a \notin \mu(x)$), so also $a{:}A \in G_{\text{ext}}$ for each such $A$. Therefore, $\{a\} \sqsubseteq k$.

(b) For $k = \bigsqcap_{(x,u):p \in P} \exists p. C_u$: If $\{a\} \sqsubseteq V_x$, then by definition $a$ is an instance of variable $x$ in $P$, i.e., $a \in \mu(x)$. Then for each property name $p$ occurring in an atomic pattern of the form $(x, u){:}p \in P$, one of two cases applies: If $u$ is an individual name, then there must be $(a, u){:}p \in G_{\text{in}}$, so also $(a, u){:}p \in G_{\text{ext}}$ for such $p$. If $u$ is a variable name, then there must be $(a, b){:}p \in G_{\text{in}}$, so also $(a, b){:}p \in G_{\text{ext}}$, and also $b \in \mu(u)$ (since otherwise $a \notin \mu(x)$). Therefore, $\{a\} \sqsubseteq k$.

(c) For $k = \bigsqcap_{(u,x):p \in P} \exists p^-. C_u$: Analogous to the previous case.

If at least one component $k$ is defined, then it follows that

$$\{a\} \sqsubseteq \bigsqcap_{x:A \in P} A \sqcap \bigsqcap_{(x,u):p \in P} \exists p. C_u \sqcap \bigsqcap_{(u,x):p \in P} \exists p^-. C_u.$$

We know, that at least one component $k$ must be defined, since otherwise concept $V_x$ would not be defined, as there must exists either $x{:}A \in P$ for some concept name $A$, or $(x, u){:}p \in P$ (or $(x, u){:}p \in P$ respectively) for some property $p$, if $x \in \text{var}(P)$. Then, at least one of the components $k$ must be defined as well, and we prove this case.

$3_{\sqsupseteq}$   *Variable ($\sqsupseteq$).* The inverse case follows directly from Lemma 1.

This concludes the proof of cases 1., 2., and 3. We next consider cases 4. and 5.

*Cases 4 and 5.* Since the proofs of these two cases are similar, we exemplify them proving the equivalency:

$$\exists \dot{p}. C_u \equiv \bigsqcup_{(v,u):p \in P} C_v$$

Let $\mathcal{I}$ be the canonical model of $G_{\text{ext}}$. By definition of the validation knowledge base of a graph, $a^{\mathcal{I}} \in (\exists \dot{p}. C_u)^{\mathcal{I}}$ if and only if there exists an individual name $b$ such that $(a, b){:}\dot{p} \in G_{\text{ext}}$ and $(a^{\mathcal{I}}, b^{\mathcal{I}}) \in p^{\mathcal{I}}$. By construction, $(a, b){:}\dot{p} \in G_{\text{ext}}$ if and only if there exists an atom $(v, u){:}p \in P$ where $v$ is the individual name $a$ or a variable $x$, and $u$ is the individual name $b$ or a variable $y$ (and thus $a^I \in C_v^{\mathcal{I}}$). Thus, $a^{\mathcal{I}} \in (\exists \dot{p}. C_u)^{\mathcal{I}}$ if and only if $a^{\mathcal{I}} \in \bigcup_{(v,u):p \in P} C_v^{\mathcal{I}}$. Hence, $\exists \dot{p}. C_u \equiv \bigsqcup_{(v,u):p \in P} C_v$.

Similarly, we exemplify the proof of the remaining axioms of the following form, using one of these axioms:

$$\exists \dot{p}. \top \equiv \bigsqcup_{(u,v):p \in P} C_u \sqcap \exists \dot{p}. C_v$$

By definition of the validation knowledge base of a graph, $a^{\mathcal{I}} \in (\exists \dot{p}. \top)^{\mathcal{I}}$ if and only if there exists an individual name $b$ such that $(a, b){:}\dot{p} \in G_{\text{ext}}$ and $(a^{\mathcal{I}}, b^{\mathcal{I}}) \in p^{\mathcal{I}}$. By construction, $(a, b){:}\dot{p} \in G_{\text{ext}}$ if and only if there exists an atom $(u, v){:}p \in P$ where $u$ is the individual name $a$ or a variable $x$ and $v$ is the individual name $b$ or a variable $y$. Then, $a^{\mathcal{I}} \in C_u^{\mathcal{I}}$ and $a^{\mathcal{I}} \in (\exists \dot{p}. C_v)^{\mathcal{I}}$, so $a^{\mathcal{I}} \in (\exists \dot{p}. \top)^{\mathcal{I}}$ if and only if $a^{\mathcal{I}} \in \bigcup_{(u,v):p \in P} C_u^{\mathcal{I}} \cap (\exists \dot{p}. C_v)^{\mathcal{I}}$.

*Conclusion.* Finally, given the proofs for the individual cases listed above, we prove this proposition.    □

## C.7   Proof for Proposition 6

We begin with the following utility definition.

**Definition 25.** For every individual name or variable $u$ we define the set of individual names $\text{ins}(u)$ as follows:

$$\text{ins}(u) = \begin{cases} \{a\} & \text{if } u \text{ is ind. name } a, \\ \{a \mid a \text{ instance of } x\} & \text{if } u \text{ is a variable } x. \end{cases}$$

In order to prove Proposition 6, we now define the following lemma, stating a relation between $\text{ins}(u)$ (Definition 25) and $C_u$ (Definition 14).

**Lemma 5.** *Let $H \leftarrow P$ be a SCCQ, $G_{\text{ext}}$ an extended graph for the query, and $u$ and $v$ two individual names or variables occurring in the query. Then, $\text{ins}(v) \subseteq \text{ins}(u)$ if and only if the inclusion $C_v \sqsubseteq C_u$ is valid for graph $G_{\text{ext}}$.*

PROOF. Let $I$ be an interpretation of the validation knowledge base of graph $G_{\text{ext}}$. By definition, for every individual $b \in \top^I$ there exists a unique individual name $a$ in the graph such that $a^I = b$. Let $u$ and $v$ be two individual or variable names occurring in pattern $P$. Then, $\cdot^I$ defines a bijection between sets $\text{ins}(u)$ and $C_u^I$, and a bijection between sets $\text{ins}(v)$ and $C_v^I$. Thus, $\text{ins}(v) \subseteq \text{ins}(u)$ if and only if $C_v^I \subseteq C_u^I$. Hence, $\text{ins}(v) \subseteq \text{ins}(u)$ if and only if $C_v \sqsubseteq C_u$. □

PROOF FOR PROPOSITION 6. Let $P_1$ and $P_2$ be components of graph pattern $P$, the function $h : \text{var}(P_1) \rightarrow \text{var}(P_2)$ be a component map, and $x$ and $y$ be two variables in $P_1$ and $P_2$, respectively, such that $h(x) = y$. According to Lemma 5, to prove that $V_y \sqsubseteq V_x$ is valid in graph $G_{\text{ext}}$ it suffices to prove $a \in \text{ins}(y)$ implies $a \in \text{ins}(x)$ for every individual name $a$.

Let $a$ be an individual name in $\text{ins}(y)$. Then, there exists a valuation $\mu$ such that $\mu(P) \subseteq G_{\text{in}}$. Let $h' : \text{var}(P) \rightarrow \text{var}(P)$ be the function that extends $h$ for the variables in $P$ that are not in the domain of $h$ as follows:

$$h'(z) = \begin{cases} z & \text{if } z \notin \text{dom}(h), \\ h(z) & \text{if } z \in \text{dom}(h). \end{cases}$$

By construction, $h'(P) \subseteq P$. Applying $\mu$ to both sides of the inclusion we get $\mu(h'(P)) \subseteq \mu(P)$. By transitivity, $\mu(h'(P)) \subseteq G_{\text{in}}$. That is, $\mu'(P) \subseteq G_{\text{in}}$ where $\mu'$ is the composite valuation $h'\mu$. Since $\mu'(x) = a$, we conclude that $a \in \text{ins}(x)$. □

## C.8 Proof for Proposition 7

PROOF. To prove Proposition 7, it suffices to show that the extension approach is sound, i.e., that both the extended and non-extended components are equivalent with respect to the bindings for all actual query variables, since then the proof for Proposition 6 applies.

Consider the variable $x$ as a target of shape $s = \psi \sqsubseteq \phi$. Then, the following extensions $\text{extp}(x, \phi)$ are permitted, depending on $\phi$:

(1) $\phi = A$ and $\{x{:}A\}$. For any input graph $G_{\text{in}}$ it holds that $\text{valid}(G_{\text{in}}, \{\psi \sqsubseteq A\})$. Then $\forall a \in \mu(x) : a{:}A \in G_{\text{in}}$, since $x$ is a target of $s$. Therefore, pattern $x{:}A$ is satisfied for all $G_{\text{in}}$.

(2) $\phi = \exists p.A$ and $\{(x, x_0){:}p, x_0{:}A\}$. For every input graph $G_{\text{in}}$, $\text{valid}(G_{\text{in}}, \{\psi \sqsubseteq \exists p.A\})$. Then for all $a \in \mu(x)$, $(a, b){:}p, b{:}A \in G_{\text{in}}$, since $x$ is a target of $s$. Therefore, patterns $(x, x_0){:}p$ and $x_0{:}A$ are satisfied for all $G_{\text{in}}$.

(3) $\phi = \exists p^-.A$ and $\{(x_0, x){:}p, x_0{:}A\}$. This case is similar to the previous case.

(4) $\phi = \forall p.A$ and $\{y{:}A \mid (x, y){:}p \in P_{\text{ext}}\}$. For every input graph $G_{\text{in}}$, $\text{valid}(G_{\text{in}}, \{\psi \sqsubseteq \forall p.A\})$. Then, for all $a \in \mu(x)$, $(a, b){:}p \in G_{\text{in}}$ implies $b{:}A \in G_{\text{in}}$, since $x$ is a target of $s$. Therefore, for any pattern $(x, y){:}p \in P_{\text{ext}}$, $y{:}A$ is satisfied for all $G_{\text{in}}$.

(5) $\phi = \forall p^-.A$ and $\{y{:}A \mid (y, x){:}p \in P_{\text{ext}}\}$. This case is similar to the previous case.

□

## C.9 Proof for Proposition 8

We separately prove the two components (1) and (2) of Definition 24 involved in Proposition 8. To this end, we write $\text{valid}(G_{\text{ext}}, \text{RS}_1(q))$ and $\text{valid}(G_{\text{ext}}, \text{RS}_2(q))$, where $q = H \leftarrow P$ is a SCCQ.

PROOF FOR $\text{valid}(G_{\text{ext}}, \text{RS}_1(q))$. For an arbitrary property name $p \in \text{voc}(P)$, the axiom $\dot{p} \sqsubseteq p$ is always true since $G_{\text{med}} \subseteq G_{\text{in}}$, by definition. When the pattern only contains $(x, y){:}p$ such that $x$ and $y$ do not occur in any other atomic patterns in $P$ (i.e., $x$ and $y$ are otherwise unrestricted), then for any $(a, b){:}p \in G_{\text{in}}$, $(a, b){:}\dot{p} \in G_{\text{med}}$. Therefore $p \sqsubseteq \dot{p}$. □

PROOF FOR $\text{valid}(G_{\text{ext}}, \text{RS}_2(q))$. Let $p \in \text{voc}(P)$ and $r \in \text{voc}(H)$, such that $P$ contains the atomic pattern $(x, y){:}p$ and $H$ contains $(x, y){:}r$, and neither $H$ nor $P$ contains any other atomic patterns involving $x$ or $y$, and $p$ or $r$, respectively. Then, for any $(a, b){:}\dot{p} \in G_{\text{med}}$ we construct $(a, b){:}\ddot{r} \in G_{\text{out}}$, therefore, $\dot{p} \sqsubseteq \ddot{r}$. In addition, since $r$ does not occur again in $H$, $\dot{p} \equiv \ddot{r}$, i.e. also $\ddot{r} \sqsubseteq \dot{p}$. □

## C.10 Theorem 1 and Proof

**Theorem 1.** *Problem IsOutputShape is NP-hard.*

PROOF. We next show that the simple graph entailment problem described by Gutierrez et al. [14] (called SGE in what follows) can be reduced to problem IsOutputShape. Problem SGE is equivalent to deciding if for a pattern $P$ consisting of two components $P_1$ and $P_2$ there is a component map $h$ from $P_1$ to $P_2$. Let $S_{\text{in}}$ be an empty set, and $q$ be the SCCQ $H \leftarrow P$ where $H$ contains an atom $u{:}A_u$ for each variable or individual name in $P$. By Proposition 6, there exists such a mapping $h$ if and only if IsOutputShape$(S_{\text{in}}, q, A_u \sqsubseteq A_x) = \text{YES}$, for each pair $(x, u)$ where $h(x) = u$ and $x$ is a variable in $P_1$ and $u$ is a variable in $P_2$. Since the number of pairs $(x, u)$ is quadratic on the size of $P$, we have shown a reduction from problem SGE to problem IsOutputShape. Since SGE is NP-hard, problem IsOutputShape is also NP-hard. □

## C.11 Proposition 9 and Proof

**Proposition 9.** *If $\text{voc}(q)$ contains $n$ concept names, and $m$ role names, then we need to iterate over $n + 2m$ target queries, and $n + 4nm + 2m$ shape constraints, and return $(n + 2m)(n + 4nm + 2m) - n$ many relevant shapes.*

PROOF. We have $n$ possible target queries with a concept name ($A$ for each $A \in \text{voc}(q)$), and $2m$ with a role name ($\exists p.\top$ and $\exists p^-.\top$ for each $p \in \text{voc}(q)$). Similarly, we have $n$ possible shape constraints including only a concept name, and $4nm$ possible shape constraints including a concept name and a role name ($\exists p.A$, $\exists p^-.A$, $\forall p.A$, and $\forall p^-.A$ for each $A \in \text{voc}(q)$ and $p \in \text{voc}(q)$). We also have $2m$ representatives for families of the form $\forall p.B$ and $\forall p^-.B$ for each

$p \in \text{voc}(q)$ and for some proxy concept name $B \notin \text{voc}(q)$. The subtrahend in the number of relevant shapes indicates the number of tautologies of the form $A \sqsubseteq A$ for all $A \in \text{voc}(q)$. $\qquad\square$

## D EXTENDING THE METHOD

We next show, how our approach can be extended to arbitrary $\mathcal{ALCHOI}$ axioms as shape constraints, and thus a much more extensive subset of SHACL. To this end, we define $\mathcal{ALCHOI}$ SHACL shapes as follows.

**Definition 26** ($\mathcal{ALCHOI}$ SHACL Syntax). *A $\mathcal{ALCHOI}$ SHACL shape is an $\mathcal{ALCHOI}$ axiom $\psi \sqsubseteq \phi$ such that the concept expressions $\phi$ is an arbitrary $\mathcal{ALCHOI}$ axiom, and $\psi$ is defined by:*

$$\psi ::= A \mid \exists p.\top \mid \exists p^-.\top$$

*A $\mathcal{ALCHOI}$ SHACL schema $S$ is an $\mathcal{ALCHOI}$ TBox that consists of a finite set of $\mathcal{ALCHOI}$ SHACL shapes.*

**Definition 27** ($\mathcal{ALCHOI}$ SHACL Semantics). *A graph $G$ is valid for a set $S$ of $\mathcal{ALCHOI}$ SHACL shapes, denoted valid($G, S$), if and only if $G$ is proof-valid according to $S$.*

We omit explicitly redefining the remainder of the main paper in terms of $\mathcal{ALCHOI}$ SHACL shapes, for the sake of simplicity, since definitions do not substantially change. Instead, we instruct the reader to consider $\mathcal{S}_{\text{in}}$, $\mathcal{S}_{\text{can}}$ and $\mathcal{S}_{\text{out}}$ (and other sets of shapes) as a set of $\mathcal{ALCHOI}$ SHACL shapes for the remainder of this section, and with respect to prior definitions.

We first consider soundness. In the remainder of this section, we present further notes on extending the method, first considering and extended axiom inferences then additional features beyond $\mathcal{ALCHOI}$ axioms. Finally, we remark how to extend the implementation.

### D.1 Soundness

In this subsection, we argue for the soundness of our presented approach for more general $\mathcal{ALCHOI}$ SHACL shapes. Indeed, we show that soundness of the method introduced in the main body of the paper is not affected by more general $\mathcal{ALCHOI}$ input shapes or shape candidates. Here, we revisit each proposition from the main body of the paper and consider, whether the proposition or its proof need to be adapted. The following section, Appendix E, gives the respective extended proofs where required.

(1) Proposition 1 is independent of the subset of SHACL, so the same proof applies.
(2) Proposition 2 must be extended for the extended set of SHACL shapes to demonstrate that the method is useful for $\mathcal{ALCHOI}$ SHACL shapes (i.e., there is a meaningful finite set of candidates), though this does not effect soundness. (See Appendix E for the extended proof.)
(3) Proposition 3 (and Corollary 1) were already proven for arbitrary $\mathcal{ALCHOI}$ axioms, and thus apply in the context of $\mathcal{ALCHOI}$ SHACL shapes as well.
(4) For Proposition 4, neither its definition, nor the definition of the UNA for a simple RDF graph, depend on the subset of SHACL.
(5) Similarly, for Proposition 5 the proof is independent of the subset of SHACL and still applies as well.

(6) The proof for Proposition 6 does not depend on the subset of SHACL shapes as well, and thus the proposition holds.
(7) Proposition 7 does involve the set of input shapes. Here, we need to decide whether we extend the approach. As per the argument in the following subsection, we consider this extension to be future work, and limit expansion to the subset of Simple SHACL shapes. Then, the proof applies and the proposition holds. Note, that this is only a minor restriction, since extending the query with respect to $\mathcal{ALCHOI}$ SHACL shapes would be limited to shapes expressible as SCCQ anyways, which essentially means that we would need to include intersection of constraints in shapes as two sets of extension patterns, which is a trivial extension to the method we present.
(8) Finally, for Proposition 8, while this step of the approach does depend on the input shapes, it only considers the role names in the vocabulary of the input shapes. Neither the method itself, nor the proof of this proposition, depend on the types of constraints expressed as input shapes, but rather are about the query patterns used to restrict these specific role names. Thus, the proposition (and its proof) still apply without modification.

Thus, the method remains sound for $\mathcal{ALCHOI}$ SHACL shapes. We only need to show that there is a sensible finite set of candidates for $\mathcal{ALCHOI}$ SHACL as well (see Appendix E for the proof). If so, then the algorithm can obtain a finite set of result shapes.

For the sake of completeness, we mention that Proposition 9 (count of candidate shapes) does no longer apply for the extended method. However, Proposition 9 is not required for soundness; indeed, a similar proposition could be formulated for the count of candidates for the extended method.

Finally, Theorem 1 (NP-hardness) clearly holds for the extended method, since the original problem can be trivially reduced to the extended problem by restricting the set of shapes to Simple SHACL.

### D.2 Extended Axiom Inference

In the previous section (and together with Appendix E), we show that our approach is sound for $\mathcal{ALCHOI}$ SHACL. This extension also makes the approach more powerful in multiple ways. The set of input shapes can be extended to $\mathcal{ALCHOI}$ SHACL, thus, more expressive constraints are considered to hold on the input graph. Similarly, the set of candidates (and thus output shapes) also includes more general shapes.

The remaining question to consider, is whether or not more axioms can now be inferred in order to improve the method: The axioms inferred for the UNA-encoding and CWA-encoding are independent of the subset of SHACL, but rather depend on the query language (and indeed graph model) used. Similarly, the axioms inferred as subsumptions between query variables (*mappings*) again depend on the query language, not on the subset of SHACL; however, the extension approach utilizing input shapes does depend on the set of input shapes. Here, a few additional rules could be added for extending the approach - we leave this as future work, since this is not a substantial addition to the method, since, again, the

actual extensions depend on shape constraints that can be translated to conjunctive CONSTRUCT queries, which is, essentially, intersection.

### D.3 SHACL Features Beyond $\mathcal{ALCHOI}$

We consider now in passing our intuition, on whether our approach can be extended for additional SHACL features.

- Node target queries. Node target queries where omitted for the sake of simplicity, since inferring such shapes based on the query template would not be very productive. We believe this is a trivial extension to our method, if a use case were to require such shapes.
- Qualified number restrictions. Using an underlying DL with support for qualified number restrictions, we believe that there would not be an issue with supporting them. However, we omit them, since we think that there are only exceedingly rare circumstances, where meaningful qualified number restriction (other than existential and universal quantification) could indeed be inferred for SPARQL CONSTRUCT queries. The particular restrictions to consider likely form a finite set informed by the query template.
- Non-cyclic shape references. Non-cyclic shape references are syntactic sugar and can be resolved by substitution. Thus, our method essentially supports non-cyclic shape references already.
- Cyclic (i.e., recursive) shape references are not supported. For recursive SHACL shapes, sets of results shapes would no longer be independent, and thus, our filtering method not applicable. However, we think that only validating a set of given shapes that include recursive shape references over the axioms inferred by our method should be possible.
- SHACL features validating literal values. We omit literals for the sake of simplicity; from SCCQ queries, no interesting constraints on literal values could be inferred. Literal value constraints that occur in the input shapes could perhaps be maintained through an encoding via some utility concept definition.

### D.4 Implementation Remarks

In order to efficiently explore the finite, but larger, search space of candidate $\mathcal{ALCHOI}$ SHACL shapes, we suggest the following approaches. Importantly, one can notice that full exploration of the candidate space is rarely required. Indeed, a subset of result shapes entailing all other shapes is most sensible for the majority of use cases, no matter whether the use case is informing users (including redundant shapes would not be necessary, but rather confusing), suggesting shapes for some data set (e.g., data integration use case), or for validation in a programming language context, or any other automatic processing of result shapes, where a minimal set entailing a larger set would generally suffice. Indeed, in such automatic cases one may not need to instantiate any shapes, but instead rely only on the set of axioms, which already can be used to check for entailment of individual shapes to the extend required by such a use case.

In order to reduce the set of candidates, one can reduce the syntax by relying on the set semantics of the set of result shapes. For example, union and intersection of constraints is not required

on the top-level of a constraint, since both can be reconstructed from entailment in the result set (e.g., if both $\psi_1 \sqsubseteq \phi_1$ and $\psi_1 \sqsubseteq \phi_2$ are result shapes, trivially, $\psi_1 \sqsubseteq \phi_1 \sqcap \phi_2$ is as well. This holds similarly for union and some types of quantification.

Secondly, one can systematically cover the search space with a breath-first search strategy, where immediate candidates are validated, before constructing more complex shapes. For example, if $\psi_1 \sqsubseteq \phi_1$ is not a result shape, then for target $\psi_1$ we do not need to validate any intersection involving $\phi_1$.

## E PROOFS (EXTENSION)

We show here, that a finite set of candidate shapes can be constructed for the extended method. To this end, we revise Proposition 2 in Proposition 10 for $\mathcal{ALCHOI}$ shapes.

**Proposition 10.** *If a* $\mathcal{ALCHOI}$ *SHACL shape* $s = \psi \sqsubseteq \phi$ *is relevant for a SCCQ* $q$, *then* $\mathrm{voc}(s) \subseteq \mathrm{voc}(q)$.

We only consider constraints, i.e., the right hand side $\phi$ of a $\mathcal{ALCHOI}$ SHACL shape $\psi \sqsubseteq \phi$, since for $\psi$, we already show that the set of target queries is finite, given a finite vocabulary of a query $q$ (Proof of Proposition 2, Appendix C.2).

Without loss of generality, we assume that all constraints are in disjunctive normal form, without (syntactical) duplications and with components sorted according to some total order (e.g., by syntactic construct and then alphanumerically by role, concept or individual names). Thus, patterns such as $A \sqcap A$ do not occur, and $B \sqcap A$ is considered equal to $A \sqcap B$. Furthermore, we omit $\forall p.C$, since it is equivalent $\neg \exists p.\neg C$.

We define the following lemmas. The first one (Lemma 6) intuitively means, that for each concept description defined according to the grammar presented in the lemma, if the vocabulary of this description is not a subset of the vocabulary of some graph $G$, then the result is either equivalent to $\top$ or $\bot$, or the concept description can be simplified, such that the resulting concept description is in the vocabulary of $G$, or equivalent to $\top$ or $\bot$.

**Lemma 6.** *Let* $G$ *be a Simple RDF graph,* $(\mathcal{T}_G, G)$ *the validation knowledge base of* $G$, *and* $C_1$ *a concept description defined by the following grammar*

$$C_1 ::= C_2 \sqcup C_1 \mid C_2 \tag{1}$$
$$C_2 ::= C_3 \sqcap C_2 \mid C_3 \tag{2}$$
$$C_3 ::= \neg C_4 \mid C_4 \tag{3}$$
$$C_4 ::= \top \mid \bot \mid A \mid \{a\} \tag{4}$$

*where* $A$ *is a concept name and* $a$ *an individual name. Then,* $\mathrm{voc}(C_1) \nsubseteq \mathrm{voc}(G)$ *implies one of the following cases:*

1. $(\mathcal{T}_G, G) \models C_1 \equiv \top$ *or* $(\mathcal{T}_G, G) \models C_1 \equiv \bot$
2. *There exists a concept description* $(\mathcal{T}_G, G) \models C_1' \equiv C_1$, *such that either* $\mathrm{voc}(C_1') \subseteq \mathrm{voc}(G)$, *or* $\mathrm{voc}(C_1') \nsubseteq \mathrm{voc}(G)$ *and* $(\mathcal{T}_G, G) \models C_1 \equiv \top$ *or* $(\mathcal{T}_G, G) \models C_1 \equiv \bot$.

PROOF. Let $\mathcal{I}$ be a model of $(\mathcal{T}_G, G)$. Note, that according to Lemma 2, every model $\mathcal{I}$ of $(\mathcal{T}_G, G)$ is isomorphic to the canonical model of $G$.

We first consider the two trivial cases for $C_4$.

1. If $C_4$ is $\top$, then trivially $(\mathcal{T}_G, G) \models \top \equiv \top$.
2. If $C_4$ is $\bot$, then trivially $(\mathcal{T}_G, G) \models \bot \equiv \bot$.

We next consider the two remaining cases for $C_4$.

(1) If $C_4$ is $A$ and $A \notin \text{voc}(G)$, then $A^{\mathcal{I}}$ is empty. Thus, $(\mathcal{T}_G, G) \models A \equiv \bot$.

(2) If $C_4$ is $a \notin \text{voc}(G)$, then $\{a\}^{\mathcal{I}}$ is empty. Thus, $(\mathcal{T}_G, G) \models \{a\} \equiv \bot$.

(3) If $C_4$ is $\exists p.C_1$, we have the following cases:

We next consider the cases for $C_3$.

(1) If $C_3$ is $\neg C_4$ and $\text{voc}(C_4) \nsubseteq \text{voc}(G)$, then either $(\mathcal{T}_G, G) \models C_4 \equiv \bot$ (and thus $(\mathcal{T}_G, G) \models \neg C_4 \equiv \top$), or $(\mathcal{T}_G, G) \models C_4 \equiv \top$ (and thus $(\mathcal{T}_G, G) \models \neg C_4 \equiv \bot$). Thus, $(\mathcal{T}_G, G) \models C_3 \equiv \top$ or $(\mathcal{T}_G, G) \models C_3 \equiv \bot$, if $\text{voc}(C_3) \nsubseteq \text{voc}(G)$.

(2) If $C_3$ is $C_4$ and $\text{voc}(C_4) \nsubseteq \text{voc}(G)$, then, as previously shown, either $(\mathcal{T}_G, G) \models C_4 \equiv \top$ or $(\mathcal{T}_G, G) \models C_4 \equiv \bot$ and thus $(\mathcal{T}_G, G) \models C_3 \equiv \top$ or $(\mathcal{T}_G, G) \models C_3 \equiv \bot$, if $\text{voc}(C_3) \nsubseteq \text{voc}(G)$.

(3) If $C_3$ is $\neg B$ and $\text{voc}(C_3) \nsubseteq \text{voc}(G)$, then $(\mathcal{T}_G, G) \models \neg B \equiv \top$, since, by definition, there exists no $b{:}B \in G$.

We next consider the cases for $C_2$ by induction.

(1) If $C_2$ is $C_3$ and $\text{voc}(C_3) \nsubseteq \text{voc}(G)$, then, as previously shown, either $(\mathcal{T}_G, G) \models C_3 \equiv \top$ or $(\mathcal{T}_G, G) \models C_3 \equiv \bot$ and thus $(\mathcal{T}_G, G) \models C_2 \equiv \top$ or $(\mathcal{T}_G, G) \models C_2 \equiv \bot$, if $\text{voc}(C_2) \nsubseteq \text{voc}(G)$.

(2) If $C_2$ is $C_3 \sqcap C_2'$ and $\text{voc}(C_3) \nsubseteq \text{voc}(G)$, then, as previously shown, either $(\mathcal{T}_G, G) \models C_3 \equiv \top$ or $(\mathcal{T}_G, G) \models C_3 \equiv \bot$. In the first case, we can reduce the term to $C_2'$ (since $\top \sqcap C \equiv C$), and by induction, either $\text{voc}(C_2') \nsubseteq \text{voc}(G)$ and then $(\mathcal{T}_G, G) \models C_2' \equiv \top$ or $(\mathcal{T}_G, G) \models C_2' \equiv \bot$, or $\text{voc}(C_2') \subseteq \text{voc}(G)$. In the second case, then also $(\mathcal{T}_G, G) \models C_2 \equiv \bot$, since $(\bot \sqcap C \equiv \bot)$.

(3) If $C_2$ is $C_3 \sqcap C_2'$ and $\text{voc}(C_3) \subseteq \text{voc}(G)$, then, for $C_2'$ if $\text{voc}(C_2') \nsubseteq \text{voc}(G)$ one of the other cases applies recursively.

We finally consider the cases for $C_1$ by induction.

(1) If $C_1$ is $C_2$ and $\text{voc}(C_2) \nsubseteq \text{voc}(G)$, then, as previously shown, either $(\mathcal{T}_G, G) \models C_2 \equiv \top$ or $(\mathcal{T}_G, G) \models C_2 \equiv \bot$ and thus $(\mathcal{T}_G, G) \models C_1 \equiv \top$ or $(\mathcal{T}_G, G) \models C_1 \equiv \bot$, if $\text{voc}(C_1) \nsubseteq \text{voc}(G)$.

(2) If $C_1$ is $C_2 \sqcap C_1'$ and $\text{voc}(C_2) \nsubseteq \text{voc}(G)$, then either, as previously shown, $(\mathcal{T}_G, G) \models C_2 \equiv \top$ or $(\mathcal{T}_G, G) \models C_2 \equiv \bot$. In the first case, then also $(\mathcal{T}_G, G) \models C_1 \equiv \top$, since $\top \sqcup C \equiv \top$. In the second case, we can reduce the term to $C_1'$ (since $\bot \sqcup C \equiv C$), and by induction, either $\text{voc}(C_1') \nsubseteq \text{voc}(G)$ and then $(\mathcal{T}_G, G) \models C_1' \equiv \top$ or $(\mathcal{T}_G, G) \models C_1' \equiv \bot$, or $\text{voc}(C_1') \subseteq \text{voc}(G)$.

(3) If $C_1$ is $C_2 \sqcup C_1'$ and $\text{voc}(C_2) \subseteq \text{voc}(G)$, then for $C_1'$ if $\text{voc}(C_1') \nsubseteq \text{voc}(G)$ one of the other cases applies recursively.

Hence, we prove the lemma. □

For Lemma 7, we slightly adapt the allowed concept descriptions by allowing existential quantification for $C_4$, thus, the concept descriptions now cover arbitrary $\mathcal{ALCHOI}$ concept descriptions in disjunctive normal form.

**Lemma 7.** *Let $G$ be a Simple RDF graph, $(\mathcal{T}_G, G)$ the validation knowledge base of $G$, and $C_5$ a concept description defined by the following grammar*

$$C_5 ::= C_6 \sqcup C_5 \mid C_6 \tag{5}$$

$$C_6 ::= C_7 \sqcap C_6 \mid C_7 \tag{6}$$

$$C_7 ::= \neg C_8 \mid C_8 \tag{7}$$

$$C_8 ::= \top \mid \bot \mid A \mid \{a\} \mid \exists p.C_5 \mid \exists p^-.C_5 \tag{8}$$

*where $A$ is a concept name and $a$ an individual name. Then, $\text{voc}(C_5) \nsubseteq \text{voc}(G)$ implies one of the following cases:*

(1) *$(\mathcal{T}_G, G) \models C_5 \equiv \top$ or $(\mathcal{T}_G, G) \models C_5 \equiv \bot$*

(2) *There exists a concept description $(\mathcal{T}_G, G) \models C_5' \equiv C_5$, such that either $\text{voc}(C_5') \subseteq \text{voc}(G)$, or $\text{voc}(C_5') \nsubseteq \text{voc}(G)$ and $(\mathcal{T}_G, G) \models C_5 \equiv \top$ or $(\mathcal{T}_G, G) \models C_5 \equiv \bot$.*

PROOF. We prove this property by induction on the structure of $C_5$ (Lemma 7). To this end, we consider first as a base cases the case where in $C_8$ existential quantification is restricted to $\exists p.C_1$ (or $\exists p^-.C_1$, respectively). According to Lemma 6, then the property under investigation holds for $C_1$. Furthermore, we assume without loss of generality, that $C_1$ is fully reduced according to Lemma 6. Thus, if $\text{voc}(C_1) \nsubseteq \text{voc}(G)$ then either $(\mathcal{T}_G, G) \models C_1 \equiv \top$ or $(\mathcal{T}_G, G) \models C_1 \equiv \bot$, since otherwise $C_1$ would not be fully reduced according to Lemma 6.

We have the following cases if $C_8$ is $\exists p.C_1$ (cases for $\exists p^-.C_1$ work exactly equivalently):

(1) If $p \notin \text{voc}(G)$, then $p^{\mathcal{I}}$ is empty, from which we can follow that $(\mathcal{T}_G, G) \models \exists p.C_1 \equiv \bot$. Note, that this holds independently from $C_1$.

(2) If $p \in \text{voc}(G)$ and $\text{voc}(C_1) \nsubseteq \text{voc}(G)$, then, by definition, either $(\mathcal{T}_G, G) \models C_1 \equiv \top$ or $(\mathcal{T}_G, G) \models C_1 \equiv \bot$. Thus, we can reduce the expression to $\exists p.\top$ or $\exists p.\bot$, respectively.

Since the cases for $C_7$, $C_6$ and $C_5$ depend only on the common property between Lemma 7 and Lemma 6, the proofs work exactly analogously to the proofs of $C_3$, $C_2$ and $C_1$ (Lemma 6), and are omitted for brevity here.

Then, by induction, starting form the restricted $C_8$ as the base case, the property follows for arbitrary concept descriptions $C_5$. Thus, we prove the lemma. □

Finally, we prove Proposition 10.

PROOF OF PROPOSITION 10. Let $s = \psi \sqsubseteq \phi$ be a $\mathcal{ALCHOI}$ SHACL shape, $q$ a SCCQ, and $G$ be a Simple RDF graph with $\text{voc}(G) \subseteq \text{voc}(q)$, and $(\mathcal{T}_G, G)$ be the validation knowledge base of graph $G$. We prove first property (i) of Proposition 10. We have the following disjoint cases:

(1) Case $\text{voc}(\psi) \nsubseteq \text{voc}(q)$. Then, by Lemma 3, $(\mathcal{T}_G, G) \models \psi \equiv \bot$ (since $\psi$ is, per definition, restricted to one of the cases covered in the lemma). Hence, shape $\psi \sqsubseteq \phi$ is not relevant (Definition 11).

(2) Case $\text{voc}(\psi) \subseteq \text{voc}(q)$ and $\text{voc}(\phi) \nsubseteq \text{voc}(q)$. Let us assume, without loss of generality, that the concept description $\phi$ is fully reduced according to Lemma 7. (Note, that if due to reduction $\text{voc}(phi) \nsubseteq \text{voc}(G)$ no longer applies, the next case below would be applicable.)

Then, according to Lemma 7, either $(\mathcal{T}_G, G) \models C_1 \equiv \top$ or $(\mathcal{T}_G, G) \models C_1 \equiv \bot$, in which case the shape is not relevant (Definition 11).

(3) Case $voc(\psi) \subseteq voc(q)$ and $voc(\phi) \subseteq voc(q)$. In this case, property (i) is trivially satisfied.

Thus we prove Proposition 10. □

**Corollary 2.** *As corollary of Proposition 10, the set of $\mathcal{ALCHOI}$ shapes over $voc(q)$ of a query $q$ is not finite.*

PROOF. Follows immediately by inspection of the syntax of a $\mathcal{ALCHOI}$ concept description over a finite vocabulary. □

However, we can further restrict the set of candidates to obtain a meaningful, finite set of shapes. To this end, we first define the quantification *nesting depth* as a property of an $\mathcal{ALCHOI}$ concept description.

**Definition 28.** The nesting depth $ndep(C)$ is defined as:

$$ndep(\exists p.C) := 1 + ndep(C) \tag{9}$$

$$ndep(\forall p.C) := 1 + ndep(C) \tag{10}$$

$$ndep(C_1 \sqcap C_2) := \max(ndep(C_1), ndep(C_2)) \tag{11}$$

$$ndep(C_1 \sqcup C_2) := \max(ndep(C_1), ndep(C_2)) \tag{12}$$

$$ndep(C) := 0 \quad \text{for all other cases} \tag{13}$$

**Example 15.** The nesting depth $ndep(\exists p.A)$ is $1 + 0 = 1$. The nesting depth $ndep(\forall p.A \sqcap \exists p.(B \sqcup \exists p.C))$ is $\max(1+0, 1+(\max(0, 1+0))) = 2$.

Then, we restrict the nesting depth of candidate $\mathcal{ALCHOI}$ SHACL shapes over the vocabulary $voc(q)$ of a given query $q$ to the diameter of the variable connectivity graph $vcg(q)$.

**Proposition 11.** *Given a query $q$, the set of relevant $\mathcal{ALCHOI}$ SHACL shapes over $voc(q)$ and with finite nesting depth is finite.*

PROOF. Follows immediately by inspection of the syntax of a $\mathcal{ALCHOI}$ concept description over a finite vocabulary and Proposition 10. □

## F RUNTIME EVALUATION

The runtime evaluation is based on randomly generated problems (i.e., sets of input shapes as well as queries). In this section, we give an overview of the included profiling tools of our implementation, and one example instance of such a evaluation run. Thus, the experiment discussed here shows basic feasibility of our method with synthetic data, though results on real-world data may differ.

### F.1 Overview

Our implementation includes tools for evaluating the runtime of the algorithm with randomly generated sample problems. The generation can be adapted for various parameters. Full results for the experiments described here are included with the implementation project source code. More results can be generated by executing the profiling application. To this end, see the included README.md file for the full documentation on how to execute and customize profiling.

Results can differ based on the reasoner implementation (multiple reasoners are available with our implementation) as well as

the optimization strategies. Some reasoner implementations or optimization strategies are not deterministic. (Note, that as a simple optimization strategy, our implementation can abort runs with a set timeout and retry computing results, in order to avoid unlucky models for non-deterministic reasoner optimization strategies.)

### F.2 Experimental Setup

We include full source code of our implementation, as well as the setup for this experiment, with this contribution. Thus, we refer to the project setup (in particular, build.sbt) with respect to version of the respective software, et. al.

Beyond that, we run experiments with Microsoft JDK build openjdk 17.0.7 2023-04-18 LTS on Windows 10 Pro (Version 10.0.19045), on commodity hardware (Intel i5-6600K @ 3.5GHz, 16GB RAM).

### F.3 Results of the Experiment

We define the following three sample configurations and give the number of atomic patterns per query (for template and pattern each) as well as the number of input shapes.

- SMALL 1-2 templates and patterns, 1-2 shapes.
- MEDIUM 5-7 templates and patterns, 5-7 shapes.
- LARGE 11-13 templates and patterns, 11-13 shapes

As a basis for these scenarios, we refer to the following real-world query datasets (logs), where most queries (more than 90%) have fewer than 6 or 7 patterns [5, 8], relating to our MEDIUM configuration. More than half include only on pattern, relating to our SMALL configuration. Our LARGE configuration covers outliers of very large queries (less than 1% of real-world queries).

For all samples, we draw fresh variables (per basic pattern) with a probability of 0.5 and fresh concepts or role names with a probability of 0.8, and sample property versus concept atomic patterns with a ratio of 0.3. We provide the full details on all parameters used for sampling with the implementation source code.

Note, that we generate shapes from the *vocabulary of the query*. Thus, the number of input shapes given here is not comparable to the size of usual sets of SHACL shapes in real-world datasets. That is, the sets of 1.5/6/12 shapes constrain the relatively small vocabulary of an input query rather tightly. We do not know of any empirical data on the average number of shapes in the query (pattern) vocabulary, i.e., that apply to a particular query, thus we estimate the numbers as given above.

We run and measure 5.000 samples each for the three given configurations, using a fixed seed for the random generator, and measure execution time for a single run of the algorithm per sample (after first running 100 additional samples as warmup). This experiment uses the HermiT[2] reasoner. A summary of results is given in Table 1.

Full output with input shapes, input queries, fine-grained execution metrics, as well as output shapes is included with the project source code as a CSV file and a summary report. Both the full output as well as the summary can be generated by executing the profile main method (see project documentation), that is, the full experiment can be executed with a single command.

---

[2]http://www.hermit-reasoner.com/

**Table 1: Results (average and median execution time in milliseconds without timeouts, the number of timeouts (limit: 10 minutes), as well as percentage of processing time spent on reasoning) for SMALL, MEDIUM and LARGE configurations.**

| Configuration | Average | Median | T/O | Reasoning |
|---|---|---|---|---|
| SMALL | 3 | 0 | 0 | 38,42% |
| MEDIUM | 40 | 20 | 0 | 87,11% |
| LARGE | 693 | 243 | 20 | 97,66% |

## F.4 Interpretation

We show in this experiment the basic feasibility of our method, with average and median execution times for even very large samples of less than one second. For the largest samples, few (0.4%) samples time out, with a set timeout of 10 minutes. We hypothesize that this is due to the reasoner sometimes choosing an unlucky model, where reasoning takes a very long time. Indeed, the majority of time is spent on reasoning for larger configuration (see Table 1) and by detailed inspection of the full log, this holds true for timeouts as well.

Received 20 February 2007; revised 12 March 2009; accepted 5 June 2009

