# OpenReview forum: "From Shapes to Shapes: Inferring SHACL Shapes for Results of SPARQL CONSTRUCT Queries"
_ACM.org/TheWebConf/2024/Conference — TheWebConf24 Oral_

### Official Review · Reviewer_zExR · 2023-11-21

**Novelty:** 4
**Technical Quality:** 5

**Review:**

# Overview:
The authors tackle the problem of determining the SHACL schema for and output of a SPARQL construct query over a set of graphs conforming to (another) SHACL schema. In particular, this is done for the SHACL shapes defined in the subfragment of the ALCHOI description logic, and for CONSTRUCT queries defined by a particular syntax. The main result of the paper is an algorithm for producing such a characterization of output graphs from CONSTRUCT queries via a set of SHACL shapes. The proposed algorithm is sound but not complete, and basically operates by iterating over all feasible shapes, checking for each one whether it is correct. The latter part, checking the "correctness" is the main technical development of the paper.

# Strengths:
- The paper is very well written and pleasant to read.
- The studied problem, while not completely new, is a good fit for the conference and the paper provides some nice insights into the question.
- There is a preliminary implementation that can be tested out, although the authors basically give it no space in the paper.

# Weaknesses:
- The work seems somewhat unfinished given that the main result provides a sound but not complete algorithm. At the very least the authors should argue why this is a reasonable approach. I do understand that the problem is highly non-trivial, but after reading through the paper it was difficult for me to distinguish it from other description logics papers I reviewed in the past. Perhaps a good justification of the approach and an explanation of why not having completeness of the proposed algorithm is not a huge deal would suite the paper well.
- There is virtually no justification as to why the particular restrictions were taken for the fragment of SHACL or SPARQL construct queries. I do agree that the studied fragment is reasonable, but the authors could justify their decisions a bit.
- For someone coming from the Semantic Web background the notation was a bit strange at first. I will give it to the authors that it is correct and consistent throughout the paper, but since the way we formalize SPARQL queries has been well established by now, I do not understand the reason to deviate form this. Not a major issue though.
- The implementation section seems completely out of place. From the writing it is difficult to judge whether the implementation is just a small proof of concept, or a fully general framework. What is written in the appendix does not really help with this either. I would suggest either removing this section, or expanding on it substantially.

# Recommendation:
Overall this work is quite solid. It tackles a reasonable problem that will be of interest to the audience of this track of the conference, and it is well written and executed. The main weakness would be that the solution itself falls a bit short, since the presentation just reads like the authors pushed the problem until they could and then stopped. Still, some nice insight is given into the problem at hand.

# Post Rebuttal:
I would like to thank the authors for clarifying some of my doubts. Overall, I would be quite happy to recommend this paper to be accepted.

# Specific comments:
- The assumption that C, I, R and V are finite is a bit problematic for the sake of availability of new concepts/roles/variables, etc. This should be countably infinite probably, since it does not change much (all the algorithms work in the context of the "instantiated" axioms).
- For Definition 11 a bit more explanation of the role of negative examples is needed. As it stands it is a bit confusing.
- In Definition 12 I started getting lost with all the dots. Perhaps introduce these not in text but as a formal definition?
- When Algorithm 2 is first presented the terms UNA, CWA, etc. are not understood. Perhaps announcing them and saying they will be explained in the remainder of the section would be good.

**Questions:**

- What would it take to have a complete algorithm as well?
- Perhaps the authors could comment a bit more on the extent of the implementation they have.

**Ethics Review Description:**

none detected

**Reviewer Confidence:**

3: The reviewer is confident but not certain that the evaluation is correct

**Scope:**

3: The work is somewhat relevant to the Web and to the track, and is of narrow interest to a sub-community

---

### Official Review · Reviewer_h9fn · 2023-11-23

**Novelty:** 5
**Technical Quality:** 5

**Review:**

This article proposes an algorithm to obtain the set of shapes that characterize the output graphs of CONSTRUCT queries. The resulting shapes are expressed using a subset of SHACL, and the queries are described using a subset of SPARQL. This is a limited but effective approach, as explained in the paper.

I’m not expert, but I think that the article is well written and provides a very detailed description of the proposal, including valuable material in the appendices. The problem addressed is also carefully formalised, as is the proposed algorithm, which is also implemented and evaluated in practical terms. The authors also provide a reproducibility artefact to test their approach.

**Questions:**

I have no questions.

**Reviewer Confidence:**

1: The reviewer's evaluation is an educated guess

**Scope:**

3: The work is somewhat relevant to the Web and to the track, and is of narrow interest to a sub-community

---

### Official Review · Reviewer_nThR · 2023-11-24

**Novelty:** 5
**Technical Quality:** 5

**Review:**

The paper presents a method that takes a SPARQL CONSTRUCT query and a set of SHACL shapes known as "source shapes," and produces a set of "target shapes" such that the result of evaluating the query over an RDF graph satisfying the sources shapes will yield a graph that satisfies the target shapes.

Technically, the query has to belong to a fairly restricted class (conjunctive queries, i.e. basic graph patterns) and the shapes also belong to a fairly restricted fragment of SHACL.  This allows solving the task in two steps:

1. enumerating all SHACL shapes that could possibly be relevant for this vocabulary
2. checking which of them are indeed implied by translating both the query and the two shape sets (along with conditions for closed world reasoning) into a description logic and using DL reasoning.

The first step is possible by excluding features like nesting from SHACL constraints. The second step relies to some extent on the limited expressivity of the query and constraint languages.

I consider this interesting work that is in scope for the track.  But I would want some questions answered before deciding whether this can be accepted for publication, see below.

Minor comments:
l 354: Should be YES/NO instead of YES/NA I suppose?
l 903: we prove

**Questions:**

Questions:

1. The problem description in line 325ff contains a minimality requirement.  I can find one more mention of minimality in the paper, saying that the algorithm does not in fact guarantee minimality.  Is this the case?  If so, why is it listed as part of the requirement instead of discussing it as a desirable feature that the algorithm does not have?  If the algorithm does give minimality, where is this stated and proven?

2. In the conclusion l913, it is stated that the appendix contains proofs of the properties of the method for a larger class of shapes.  Why is this stronger statement not part of the main paper?  Why lift one of the main restrictions in the last paragraphs of the paper?

3. The related work concentrates on other work concerning semantic technologies.  But the problem addressed is very related to that of inferring constraints on database views.  Related work from the relational database community should be discussed, or it should be pointed out why that work is not relevant for the problem solved here.

**Reviewer Confidence:**

4: The reviewer is certain that the evaluation is correct and very familiar with the relevant literature

**Scope:**

4: The work is relevant to the Web and to the track, and is of broad interest to the community

---

### Official Review · Reviewer_W8LY · 2023-11-26

**Novelty:** 5
**Technical Quality:** 4

**Review:**

The problem of deriving target  SHACL constraints from source SHACL constraints and simple SPARQL CONSTRUCT queries is considered. A sound (but not complete) algorithm is proposed and implemented.

The topic is certainly relevant and interesting. In fact, a similar problem for databases has been around for a very long time, and there is a considerable number of classical results, none of which are mentioned in the submission. Writing is quite heavy, and the intuition is very poorly explained - it's very difficult to get through the dotted, double-dotted symbols, with all the permutations of role inverses, etc.

Why are the sets C, I, R, and V finite? Given that they are fixed, that leaves one with a very limited setting - the number of distinct simple RDF graphs, for example, is bounded by |I|^2 x |R| + |I| x |C|. Is it needed only for Definition 4? In that case, one could restrict the sets in (1) and (2) to the individuals that occur in G (but I itself can be infinite), by using the notion of vocabulary (Definition 10).

Do grammars for \psi and \phi (Definition 5) not miss \top? It appears in Example 1.

Atomic pattern in Definition 7 looks almost like a triple pattern from the SPARQL standard - why a new term then? The term BGPs is re-used, although with minor changes again.

Definition 13 sounds very complicated - "a minimal set", the "axiom is in" - why is it not simply "UNA(q) is the set of axioms of the form ..., for distinct a and b in q?" The same applies to Definition 18 - the "minimal set .. containing" is just an unnecessary complication in what should be a very simple definition.

What are "utility functions" (l 534)? Is it just concepts?

What is the meaning of "vcg(P) is acyclic w.r.t. to[sic!] x"? By the way, "t." in "w.r.t." comes from "to".

The pairs of axioms in items 4 and 5 in Definition 16 can be written more concisely if one uses \rho rather than p with the assumption that p^-(b,a) \in P whenever p(a,b)\in P.

In Example 5, why is \subset CWA(q1) repeated in every item if the text above the items clearly says that these are in CWA(q1)?

The definition at the beginning of Section 6.2 is normally written "a maximal connected subset", without any "there exists no...".

In Definition 17, the h-image of P1 is normally denoted h(P1), without any confusing superscripts.

------

SCCQ (l 23) - does it stand for SPARQL CONSTRUCT Conjunctive Queries? The abbreviation comes out of the blue.

"fruitful composition" (l 71) is a strange phrase - meaningful?

A developer -> The developer (l 81)

The sentence "We interpret all RDF classes, etc." (l 115) should really start the paragraph - otherwise, "description logic" comes a b it of a shock. And "For clarity, " (l 122) should also appear much earlier in the text.

model*-*theoretic (l 168 and others)

ll 205-206: (a,b):p *\in G* and similarly for (b,a)

ll 230-231: why not use \rho to halve the number of options in the grammars for \psi and \phi?

l 287: a runaway . at the start of the line

l 310: \equiv \bot would look better as \sqsubseteq \bot

l 346: reflexivity  (not -ness)

**Questions:**

See the review.

**Ethics Review Description:**

-

**Reviewer Confidence:**

3: The reviewer is confident but not certain that the evaluation is correct

**Scope:**

4: The work is relevant to the Web and to the track, and is of broad interest to the community

---

### Official Review · Reviewer_Beni · 2023-11-29

**Novelty:** 5
**Technical Quality:** 6

**Review:**

The work proposes the derivation of shape constraints  for RDF graphs and SPARQL CONSTRUCT query. The proposed constraints hold
on all possible output graphs of a given SPARQL CONSTRUCT query. The paper is theoretically very solid with proofs and running examples. There also include evaluation results. Overall, the paper might be a difficult to follow for non-expert users. An online demo might be very helpful as well to allow user to apply the proposed shape constraints on different SPARQL CONSTRUCT queries.

**Questions:**

Overall the paper looks very solid. I have no specific question to ask. I believe more running examples, demo etc. could be very helpful.

**Ethics Review Description:**

Nothing

**Reviewer Confidence:**

1: The reviewer's evaluation is an educated guess

**Scope:**

3: The work is somewhat relevant to the Web and to the track, and is of narrow interest to a sub-community

---

### Decision · Program_Chairs · 2024-01-22

**Decision:**

Accept (Oral)

**Comment:**

This article introduces an algorithm to derive a set of target SHACL shapes from a SPARQL CONSTRUCT query and source SHACL shapes.

 The reviewers agree that this is novel work and that the paper is clearly written.
 As such, it was agreed that this is a valuable contribution for the Web Conference, and deserves to be accepted.
 We recommend the authors to incorporate the comments and clarifications that arose during the discussions.